# SUBSTRUCTURE-ATOM CROSS ATTENTION FOR MOLECULAR REPRESENTATION LEARNING

## ABSTRACT

Designing a neural network architecture for molecular representation is crucial for AI-driven drug discovery and molecule design. In this work, we propose a new framework for molecular representation learning. Our contribution is threefold: (a) demonstrating the usefulness of incorporating substructures to node-wise features from molecules, (b) designing two branch networks consisting of a transformer and a graph neural network so that the networks fused with asymmetric attention, and (c) not requiring heuristic features and computationally-expensive information from molecules. Using 1.8 million molecules collected from ChEMBL and PubChem database, we pretrain our network to learn a general representation of molecules with minimal supervision. The experimental results show that our pretrained network achieves competitive performance on 11 downstream tasks for molecular property prediction.

## 1 INTRODUCTION

Predicting properties of molecules is one of the fundamental concerns in various fields. For instance, researchers apply deep neural networks (DNNs) to replace expensive real-world experiments to measure the molecular properties of a drug candidate, e.g., the capability of permeating the blood-brain barrier, solubility, and affinity. Such an attempt significantly reduces wet-lab experimentation that often takes more than ten years and costs $1 million (Hughes et al., 2011; Mohs & Greig, 2017).

Among the DNN architectures, graph neural networks (GNNs) and Transformers are widely adopted to recognize graph structure of molecules. GNN architectures for molecular representation learning include message-passing neural network (MPNN) and directed MPNN (Gilmer et al., 2017; Yang et al., 2019), where they investigate how to obtain effective node, edge, and graph representation. GNNs are powerful in capturing local information of a node, but may lack the ability to encode information from far-away nodes due to over-smoothing and over-squashing issues (Li et al., 2018; Alon & Yahav, 2020). On the other hand, Transformer-based architectures such as MAT (Maziarka et al., 2020) and Graphormer (Ying et al., 2021) augment the self-attention layer of a vanilla Transformer using high-order graph connectivity information. Transformers can encode global information as they consider attention between every pair of nodes from the first layer. To guide a structural bias in the attention mechanism, previous work relies on heuristic features such as the shortest path between two nodes since the naive Transformers cannot recognize the graph structure.

From the understanding of chemical structure, it is known that meaningful substructures can be found across different molecules, also known as motif or fragments (Murray & Rees, 2009). For example, carbon rings and NO2 groups are typical substructures contributed to mutagenicity (Debnath et al., 1991) showing that proper usage of substructures can help a property prediction. Molecular substructures are often represented as molecular fingerprints or molecular fragmentation. Molecular fingerprints such as MACCS (Molecular ACCess System) keys (Durant et al., 2002) and Extended-Connectivity Fingerprints (ECFPs) Rogers & Hahn (2010) represent a molecule into a fixed binary vector where each bit indicates the presence of a certain motif in the molecule. With a predefined fragmentation dictionary, such as BRICS (Degen et al., 2008) or tree decomposition (Jin et al., 2018), a molecule can be decomposed into distinct partitions. Interestingly, machine learning algorithms that utilize molecular substructures still show competitive performance on some datasets to deep learning models (Hu et al., 2020; Maziarka et al., 2020).

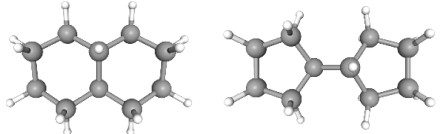 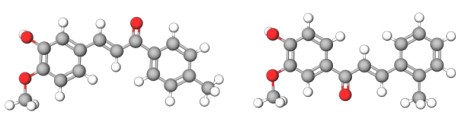

(a) Indistinguishable molecules by traditional GNNs      (b) Indistinguishable molecules by MACCS keys

Figure 1: Traditional GNNs, such as GCN and GAT, cannot distinguish the two molecular graphs in (a). However, it can be easily distinguished through simple 6-ring and 5-ring substructure information. On the other hand, the two molecules in (b) have similar substructures, so atom features from neighborhood are necessary to discriminate the two molecules.

We propose a fusion architecture between a GNN and Transformer to incorporate molecular graph information and molecular substructures. Molecular substructures and graph are encoded through Transformer and GNN, respectively. The Transformer is designed to recognize the molecular substructures, where the substructure information is mixed through self-attention to obtain better representations. With the Transformer only architecture, however, local information of molecules, such as atoms, bonds, and connectivity, can be lost from the structures. For example, the two molecules shown in Figure 1b share the same representation with MACCS keys while having different structures. To overcome, we use a separate GNN branch for preserving local information. In our model, we inject the GNN feature into the intermediate Transformer layers through the fusing network. In this way, substructures and local node information are interactively fused, producing a final representation for molecular graphs.

We name our network as ***Substructure-Atom Cross Attention (SACA)*** as it uses substructure as well as atom information in molecules and fuses them through cross-attention. The architectural choices allows us to avoid heuristic features such as the shortest-path or 3-dimensional distance in attention layers, which are required in many existing Transformer architectures for molecular representation learning (Maziarka et al., 2020; Ying et al., 2021). Furthermore, our model reduces the complexity for attention calculation over the node-level Transformer models from $O(N^2)$ to $O(N)$, where $N$ is the number of atoms. To demonstrate the empirical effectiveness of the proposed network and see the ability to capture the general representation of molecules, we evaluate our model on 11 downstream tasks from MoleculeNet (Wu et al., 2018). Our approach achieves the competitive performance on 11 downstream tasks.

In what follows, we summarize the key contributions and benefits.

- We propose a novel network that combines the information from substructures and node features in a molecule. Our model combines the advantages of both Transformer and GNN architectures to represent the information given to each architecture.
- We show the effectiveness of our model for molecular representation learning. Our model achieves competitive performance upon strong baseline models on 11 molecular property tasks.
- Our model does not require computationally-expensive heuristic information of molecular graphs.
- The source code and pretrained networks will be released in the public domain upon the paper acceptance.

## 2 RELATED WORK

### 2.1 ARCHITECTURES FOR MOLECULAR REPRESENTATION LEARNING

**Graph neural networks (GNNs)** Most common architecture for molecular representation learning is the GNNs since molecules can be naturally represented as a graph structure; a node as an atom, an edge as a connection. Researchers have actively investigated variations of GNN architectures (Gilmer et al., 2017; Yang et al., 2019; Xiong et al., 2019; Song et al., 2020), for molecules. For

example, MPNN (Gilmer et al., 2017) generalizes the message passing frameworks and explores some variants that predict molecular properties. Directed MPNN (DMPNN) (Yang et al., 2019) proposed to replace the node-based message by edge-based messages to avoid unnecessary message loops. Next, communicative MPNN (CMPNN) (Song et al., 2020) improved DMPNN by additionally considering the node-edge interaction during the message passing phase. AttentiveFP (Xiong et al., 2019) extends the graph attention mechanism to allow for nonlocal effects at the intramolecular level.

Despite the advance of GNN architectures for molecular representation learning, there are known problems in GNN, such as over-smoothing and over-squashing problems (Li et al., 2018; Alon & Yahav, 2020), which means the node representations become too similar, and the information from far nodes does not propagate well as the number of neighbors increases exponentially. Furthermore, Xu et al. (2019) analyzes expressive powers of standard GNNs following neighborhood aggregation scheme and shows that they are bounded to Weisfeiler-Lehman test (WL-test) (Weisfeiler & Leman, 1968). Therefore, GNNs with standard message passing cannot learn to discern a simple substructure such as cycles. For example, the two molecules in 1a cannot be distinguished by WL-test, hence standard GNNs cannot distinguish these two molecules (Bodnar et al., 2021). A solution to this limited representation power of GNNs is to directly incorporate important substructures in the representation learning framework.

**Transformers**    With recent advance of Transformer architectures and their promising performance in various domains, including NLP and computer vision (Devlin et al., 2019; Dosovitskiy et al., 2020), Transformer-based architectures for molecular representation learning (Maziarka et al., 2020; Ying et al., 2021; Rong et al., 2020; Maziarka et al., 2021) have been developed. Transformer architecture calculates pair-wise attention between every node from the first layer. Therefore, it can effectively capture the global information of a graph. However, a vanilla Transformer architecture (Vaswani et al., 2017) is not directly applicable for molecular graph representation because it cannot incorporate structural information such as the edge and connectivity in graphs. To bridge the gap, advanced Transformer architectures alter the self-attention layer (Maziarka et al., 2020; Ying et al., 2021; Maziarka et al., 2021) or incorporate message passing networks into Transformer architectures for input feature (Rong et al., 2020). Specifically, MAT (Maziarka et al., 2020) uses adjacency and distance matrix of atoms to augment the self-attention layer. GROVER (Rong et al., 2020) runs Dynamic Message Passing Network (dyMPN) over the input node and edge features to extract queries, keys and values for self-attention layers. CoMPT (Chen et al., 2021) uses the shortest path information between two nodes, and Graphormer (Ying et al., 2021) encodes node's degree, edges and the shortest path with edges between two nodes for molecular data. Although these graph-specific features such as the shortest path or 3D distance are found to be useful in graph representation, these features can be heuristic and impose excessive computational overhead, which limits the model's applicability to large molecules such as proteins. Our proposed model does not require any computationally expensive heuristic features such as 3D information or shortest path for preprocessing or computation in attention layer.

## 2.2 Molecular Substructure

Substructure information extracted from molecules has been widely used in molecular generation, property prediction, and virtual screening (Willett et al., 1998; Brown & Martin, 1996; Eckert & Bajorath, 2007; Jin et al., 2018). ECFPs (Rogers & Hahn, 2010) encode existing substructures within a circular distance from each atom in a molecule. PMTNN (Ramsundar et al., 2015) is a multi-task network that takes ECFPs as an input to predict molecular properties on various datasets. MACCS keys (Durant et al., 2002) extract molecules substructure depending on the presence of pre-defined functional groups. One example of the usage of MACCS keys is to encode known ligands of each protein, which leads to an improvement of prediction performance in protein-ligand interaction (Li et al., 2019). Molecular fragmentation method such as BRICS fragmentation (Degen et al., 2008; Zhang et al., 2021) or tree decomposition (Jin et al., 2018) is to decompose molecules in non-overlapping partitions with pre-defined rules. BRICS fragmentation divides molecules following chemical reaction based rules. Tree decomposition proposed in Jin et al. (2018) also extracts junction tree by contracting certain edges. One node in the junction tree represents a substructure of original molecules. Our model receives molecular substructures as input to supplement GNN's expressivity and it can flexibly encode any substructure vocabulary.

Our model combines Transformer and GNN, with molecular substructures and molecular graph as inputs to each network. There are existing studies that have also considered combining GNN

and Transformer architecture (Zhu et al., 2021a;b; Yang et al., 2021) to learn graph representations. Specifically, DMP (Zhu et al., 2021a) utilizes GNN and Transformer to encode graphs and SMILES representation of molecules and train both branches using a consistency loss to match the two outputs of input molecules. PoseGTAC (Zhu et al., 2021b) and GraphFormers (Yang et al., 2021) combine GNN and Transformer layer alternatively to enlarge the receptive field or to mix the output of Transformer. Our model is the first to encode substructures through Transformer and inject atom features through a separate GNN. In this way, we preserve both substructures and local atom features of molecules.

## 3 MODEL

In this section, we explain the architecture of our model.

**Overall architecture** Figure 2 shows the overall architecture of our model. Our network consists of two branches: (i) a Transformer branch that uses the molecular substructures as input and (ii) a GNN branch that uses a molecular graph as input. The two branches have different roles. First, the Transformer branch is intended to capture global information of molecules. It receives the molecular substructures that have important role in molecular properties, but cannot be easily captured by GNNs, and learn the overall representation of molecules. On the other hand, the GNN branch is intended to capture local node information of molecules. The two different levels of information are mixed through the fusing network in the Transformer branch.

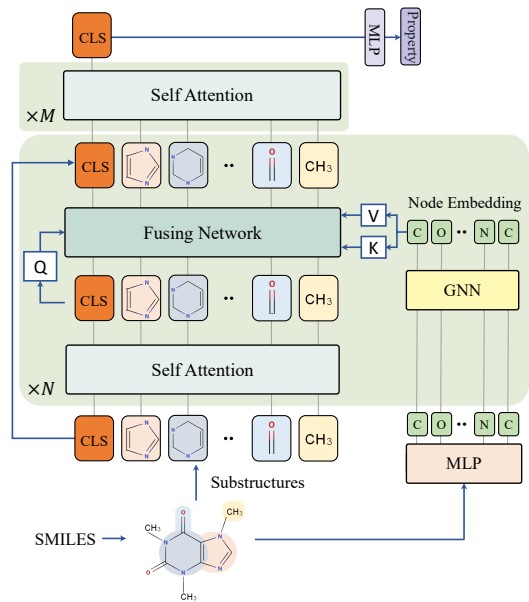

Figure 2: Overall architecture. Our model consists of Transformer and GNN branches. Transformer encodes molecular substructures, while GNN encodes atomic information. The self-attention and fusing network are alternatively stacked $N$ times, followed by $M$ more self-attention layer to refine the substructure feature. The final CLS token is used to predict the molecular properties.

**Transformer branch** Our Transformer branch is to incorporate both molecular substructure information and local node features. The input token for Transformer is the substructure embeddings of molecules. Predefined substructures are first detected and then projected into separate embedding vectors. For example, in Figure 2, substructures such as N-Heterocycle, carbon-oxygen bond and methyl group are detected and embedded into learnable embedding vectors. To identify substructure from the input molecules, we use MACCS keys (Durant et al., 2002), which indicates the presence of motifs in a molecule. Note that our architecture is not limited to certain molecular substructures, but it can flexibly receive any substructure vocabulary. The embeddings of substructures are mixed together and refined as they are passed through the self-attention module.

The substructure embeddings after self-attention layer are fused with node embeddings from a separate GNN branch. The fusing network computes cross-attention between substructures and nodes where substructures are used as query and nodes are used as key and value. The detailed computation of the fusing network is shown in Figure 3. In the fusing network, the cross-attention between each pair of substructure embedding and node embedding is computed.

To be specific, for a given molecule having $n$ atoms and $m$ extracted substructures, we have substructure embeddings $E_s \in \mathbb{R}^{m \times d}$ and node embeddings $E_n \in \mathbb{R}^{n \times d}$ where $d$ is the embedding dimension. Then, the cross-attention is computed as follows:

$$\text{Attention}(Q, K, V) = \text{Softmax}\left(\frac{(E_s W_Q)(E_n W_K)^T}{\sqrt{d_k}}\right)(E_n W_V), \qquad (1)$$

where $W_Q, W_K, W_V \in \mathbb{R}^{d \times d_k}$ are learnable parameters. The cross-attention module outputs $E_s' \in \mathbb{R}^{m \times d_k}$. Through this fusing network, the substructure embeddings aggregate the local information from node embeddings. Structurally important nodes are aggregated with more weights. Instead of designing heuristic weights on the nodes, our model can learn to select structurally important nodes related to graph-level property by cross-attention. Additionally, as the attention is computed between substructures and nodes, the space and time complexity of the self and cross attention map of our model is linear to the number of atoms, i.e., $O(N)$, whereas other transformer architectures for molecular graphs have quadratic complexity.

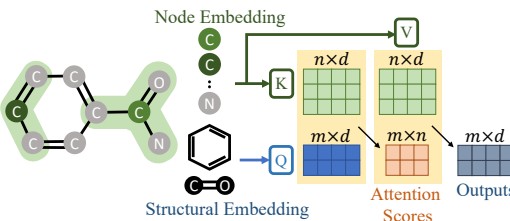

Figure 3: Illustration of structural and node embedding for cross attention computation. Cross attention is computed with the illustrated query, key and value.

The self-attention between substructures and fusing network with node embeddings are repeated iteratively $N$ times. Before making the final prediction, we add $M$ self-attention layers at the end of the network for making refinement on the substructures. We add a residual connection from the input tokens to every output of the fusing network. This ensures the input structural information last throughout the entire network. Furthermore, we use a CLS token similar to special classification token in BERT (Devlin et al., 2019) to aggregate the global representation of molecules. The CLS token is shown in Figure 2. It is a learnable latent vector and passed to the Transformer branch attached to the substructure tokens as input. A CLS token has been used in many Transformer-based architectures (Maziarka et al., 2020; Ying et al., 2021) for molecular representation learning. We attach an MLP head to the final CLS token to perform graph-level prediction tasks.

**GNN branch** The GNN branch is used to extract local node features from molecular graphs. For GNN architecture, we use GIN (Xu et al., 2019) with jumping knowledge (Xu et al., 2018). The computed node features are injected into the Transformer through the fusing network with the same hierarchy. For example, 0-hop node representations are injected into the first cross-attention module and 1-hop representations for the second cross-attention module. This allows the model to encode local node features progressively from the shallow to deeper layers.

The computation of node representation through GNNs and injection to the Transformer allow us to take advantage of both GNNs and Transformer architectures. Substructures that are hard to be captured by GNNs, but essential to molecular properties, are first detected and encoded through Transformer. Meanwhile, local node information that can be lost in using substructures alone is effectively captured by GNNs and fused with substructures. Additionally, our architecture does not require computationally expensive high-order graph-level information. As Transformer cannot naturally incorporate a graph's edge connectivity information, existing work (Rong et al., 2020; Ying et al., 2021; Chen et al., 2021) mainly focuses on how to add structural bias such as the shortest path or 3D distance between two nodes into the Transformer self-attention computation. However, these structural biases require computationally expensive preprocessing of the molecular datasets. Our network that utilizes GNN as a separate branch can avoid these limitations.

## 4 EXPERIMENT

### 4.1 EXPERIMENTAL SETTING

**Pretraining** We pretrain our network to obtain molecular representation transferable to various molecular datasets and tasks. For pretraining, we extracted 200 real-valued descriptors of physicochemical properties from the pretraining datasets using RDKit (Landrum, 2016) and train our network to predict these properties. As the 200 molecular descriptors include a diverse set of molecular properties, the model can learn a representation of molecules that can be used for various downstream tasks.

**Dataset** We collected 1,858,081 number of unlabeled molecules from ChEMBL and PubChem databases (Kim et al., 2016; Gaulton et al., 2012). We preprocessed the pretraining dataset to remove

Table 1: Statistics of eleven datasets from MoleculeNet (Wu et al., 2018) used for downstream tasks.

(a) Classification tasks

| Dataset | Size | # Tasks | Metric |
|---------|------|---------|--------|
| BBBP | 2,039 | 1 | ROC-AUC |
| SIDER | 1,427 | 27 | ROC-AUC |
| ClinTox | 1,478 | 2 | ROC-AUC |
| BACE | 1,513 | 1 | ROC-AUC |
| Tox21 | 7,831 | 12 | ROC-AUC |
| ToxCast | 8,575 | 617 | ROC-AUC |

(b) Regression tasks

| Dataset | Size | # Tasks | Metric |
|---------|------|---------|--------|
| FreeSolv | 642 | 1 | RMSE |
| ESOL | 1,128 | 1 | RMSE |
| Lipophilicity | 4,200 | 1 | RMSE |
| QM7 | 6,830 | 1 | MAE |
| QM8 | 21,786 | 12 | MAE |

duplicated molecules in the pretraining dataset and keep the downstream dataset intact. Further details of pretraining dataset collection procedure are presented in Appendix D. ChEMBL and PubChem are large-scale databases that include a variety of chemical and physical properties, and biological activities of molecules. To obtain molecular substructures, we utilize MACCS key (Durant et al., 2002), a 166 dimensional vector that indicates a presence of certain substructure in molecules, and extract this for every molecule using RDKit (Landrum, 2016). We use OGB package (Hu et al., 2020) to convert SMILES (Weininger, 1988), a text-representation for molecules, to molecular graphs.

**Implementation details**  We set $M = 4$ and $N = 3$, where $M$ and $N$ are defined in section 3. We set 768-dimensional hidden units and 16 attention heads. When pretraining, we used the AdamW optimizer (Loshchilov & Hutter, 2018) with a learning rate of 1e-4. We divide the pretraining dataset into a 9:1 ratio and use them for training and validation sets. The model is trained for 10 epochs, and the model with the best validation loss is used for downstream tasks. Further details for pretraining setting is presented in Appendix D.

## 4.2 DOWNSTREAM TASKS

**Tasks**  We evaluate the performance on six classification tasks (BBBP, SIDER, ClinTox, BACE, Tox21, and ToxCast) and five regression tasks (FreeSolv, ESOL, Lipo, QM7 and QM8) from MoleculeNet (Wu et al., 2018). The dataset statistics are summarized in Table 1. Each task is related to molecular property from low-level, for example, water solubility in ESOL to high-level, possibility of blood-brain barrier penetraion in BBBP. Further details about the downstream tasks are available in Appendix B.

**Experimental setting**  To evaluate each model, we use 3 different scaffold splits (Wu et al., 2018) following Rong et al. (2020). Scaffold split divides structurally different molecules into different subsets and provides more challenging and realistic test environment. From the pretrained model, we replace the last MLP layer of the network with the task-specific MLP heads. For each downstream dataset, we train our model for 100 epochs and report the test score corresponding to the best validation epoch. We tune hyperparameters with Bayesian optimization search with a budget of 100 for learning rate, dropout, weight decay and the number of last prediction heads. The hyperparameter search range is provided in Appendix D.

We compare the performance of our model on the downstream tasks with several GNN and Transformer based state-of-the-arts approaches for molecule representation learning. TF-Robust (Ramsundar et al., 2015) is a DNN-based model that takes molecular fingerprints. GNN-based models include GraphConv (Kipf & Welling, 2017), Weave (Kearnes et al., 2016) and SchNet (Schütt et al., 2017) which are 3 graph convolutional networks, MPNN (Gilmer et al., 2017), DMPNN (Yang et al., 2019), MGCN (Lu et al., 2019) and CMPNN (Song et al., 2020) which are GNN models considering the edge features during message passing. AttentiveFP (Xiong et al., 2019) is an extension of graph attention network for molecule representation. Transformer-based models include GROVER (Rong et al., 2020), MAT (Maziarka et al., 2020), Graphormer (Ying et al., 2021) and CoMPT (Chen et al., 2021). Among the baselines, N-GRAM (Liu et al., 2019), Hu et al. (2019), GraphLoG (Xu et al., 2021), MAT, GROVER, Graphormer, GEM (Fang et al., 2022) and MPG (Li et al., 2021) are models that use pretraining strategies. We report GROVER base model for a fair comparison in terms of the number of parameters. To reproduce the results for models using pretraining strategies, we use the pretrained model made available by the authors. We reproduce the results of MPG due to the

Table 2: Comparison on small-scale datasets. We report the average and standard deviation (in brackets) over three splits. We mark the best and the second-best performances in **bold yellow** and light yellow, respectively.

### Classification Tasks

| Method | Pre. | BBBP ↑ | SIDER ↑ | ClinTox ↑ | BACE ↑ | Tox21 ↑ | ToxCast ↑ | Rank |
|---|---|---|---|---|---|---|---|---|
| TF_Robust (Ramsundar et al., 2015) | - | .860(.087) | .607(.033) | .765(.085) | .824(.022) | .698(.012) | .585(.031) | 17.5 |
| Weave (Kearnes et al., 2016) | - | .837(.065) | .543(.034) | .823(.023) | .791(.008) | .741(.044) | .678(.024) | 18.5 |
| GraphConv (Kipf & Welling, 2017) | - | .877(.036) | .593(.035) | .845(.051) | .854(.011) | .772(.041) | .650(.025) | 15.2 |
| SchNet (Schütt et al., 2017) | - | .847(.024) | .545(.038) | .717(.042) | .750(.033) | .767(.025) | .679(.021) | 19.2 |
| MPNN (Gilmer et al., 2017) | - | .913(.041) | .595(.030) | .879(.054) | .815(.044) | .808(.024) | .691(.013) | 12.7 |
| DMPNN (Yang et al., 2019) | - | .919(.030) | .632(.023) | .897(.040) | .852(.053) | .826(.023) | .718(.011) | 6.0 |
| MGCN (Lu et al., 2019) | - | .850(.064) | .552(.018) | .634(.042) | .734(.030) | .707(.016) | .663(.009) | 20.2 |
| AttentiveFP (Xiong et al., 2019) | - | .908(.050) | .605(.060) | .933(.020) | .863(.015) | .807(.020) | .579(.001) | 11.5 |
| CMPNN (Song et al., 2020) | - | .940(.009) | .612(.006) | .931(.003) | .868(.033) | .805(.017) | .722(.005) | 5.5 |
| CoMPT (Chen et al., 2021) | - | .930(.019) | .605(.011) | .818(.081) | .851(.043) | .790(.031) | .716(.010) | 10.2 |
| Molformer (Wu et al., 2021b) | - | .926(.037) | .616(.015) | .674(.054) | .842(.024) | .773(.013) | .688(.008) | 13.2 |
| N-GRAM (Liu et al., 2019) | ✓ | .912(.013) | .632(.005) | .855(.037) | .876(.035) | .769(.027) | - | 9.4 |
| Hu. et.al (Hu et al., 2019) | ✓ | .915(.040) | .614(.006) | .762(.058) | .851(.027) | .811(.015) | .714(.019) | 10.2 |
| MAT (Maziarka et al., 2020) | ✓ | .922(.035) | .617(.012) | .853(.079) | .830(.045) | .810(.015) | .712(.004) | 9.3 |
| GROVER (Rong et al., 2020) | ✓ | .936(.008) | .656(.006) | .925(.024) | .878(.016) | .819(.020) | .723(.010) | 2.3 |
| Graphormer (Ying et al., 2021) | ✓ | .938(.032) | .625(.009) | .913(.056) | .848(.024) | .801(.013) | .718(.007) | 6.8 |
| GraphLoG (Xu et al., 2021) | ✓ | .913(.024) | .595(.039) | .749(.050) | .845(.012) | .773(.010) | .677(.008) | 15.5 |
| GraphMVP (Liu et al., 2021) | ✓ | .923(.032) | .617(.023) | .740(.096) | .850(.022) | .788(.023) | .694(.022) | 11.5 |
| MGSSL (Zhang et al., 2021) | ✓ | .912(.023) | .616(.026) | .763(.031) | .847(.037) | .795(.015) | .695(.013) | 12.5 |
| MPG* (Li et al., 2021) | ✓ | .922(.039) | .628(.014) | - | .864(.028) | .800(.024) | .712(.009) | 7.8 |
| GEM (Fang et al., 2022) | ✓ | .921(.026) | .603(.012) | - | .872(.036) | .815(.016) | .720(.010) | 7.6 |
| Ours | ✓ | .934(.018) | .646(.009) | .935(.014) | .877(.032) | .829(.013) | .730(.005) | **1.8** |

### Regression Tasks

| Method | Pre. | FreeSolv ↓ | ESOL ↓ | Lipo ↓ | QM7 ↓ | QM8 ↓ | Rank |
|---|---|---|---|---|---|---|---|
| TF_Robust (Ramsundar et al., 2015) | - | 4.122(.085) | 1.722(.038) | .909(.060) | 120.6(9.6) | .024(.001) | 15.2 |
| Weave (Kearnes et al., 2016) | - | 2.398(.250) | 1.158(.055) | .813(.042) | 94.7(2.7) | .022(.001) | 11.8 |
| GraphConv (Kipf & Welling, 2017) | - | 2.900(.135) | 1.068(.050) | .712(.049) | 118.9(20.2) | .021(.001) | 12.2 |
| SchNet (Schütt et al., 2017) | - | 3.215(.755) | 1.045(.064) | .909(.098) | 74.2(6.0) | .020(.002) | 11.4 |
| MPNN (Gilmer et al., 2017) | - | 2.185(.952) | 1.167(.430) | .672(.051) | 113.0(17.2) | .015(.002) | 10.0 |
| DMPNN (Yang et al., 2019) | - | 2.177(.914) | .980(.258) | .653(.046) | 105.8(13.2) | .0143(.002) | 7.8 |
| MGCN (Lu et al., 2019) | - | 3.349(.097) | 1.266(.147) | 1.113(.041) | 77.6(4.7) | .022(.002) | 13.6 |
| AttentiveFP (Xiong et al., 2019) | - | 2.030(.420) | .853(.060) | .650(.030) | 126.7(4.0) | .0282(.001) | 8.8 |
| CMPNN (Song et al., 2020) | - | 2.254(.356) | .841(.090) | .606(.040) | 70.6(2.7) | .0136(.001) | 4.8 |
| CoMPT (Chen et al., 2021) | - | 2.125(.590) | .898(.053) | .632(.038) | 65.3(3.4) | .0145(.002) | 5.6 |
| N-GRAM (Liu et al., 2019) | ✓ | 2.512(.190) | 1.100(.160) | .876(.033) | 125.6(1.5) | .0320(.003) | 14.0 |
| MAT (Maziarka et al., 2020) | ✓ | 2.116(.152) | .833(.122) | .668(.025) | 93.6(13.8) | .0178(.002) | 6.6 |
| GROVER (Rong et al., 2020) | ✓ | 1.592(.072) | .888(.116) | .563(.030) | 72.5(5.9) | .0172(.002) | 4.4 |
| Graphormer (Ying et al., 2021) | ✓ | 2.089(.150) | .827(.086) | .674(.035) | 171.3(10.7) | .0140(.002) | 7.2 |
| MPG* (Li et al., 2021) | ✓ | 2.44(.520) | .926(.159) | .682(.013) | - | - | 10.7 |
| GEM (Fang et al., 2022) | ✓ | 2.21(.374) | .885(.115) | .617(.023) | 58.9(3.9) | .0135(.001) | 4.4 |
| Ours | ✓ | 1.750(.170) | .822(.073) | .575(.009) | 63.5(3.4) | .0134(.002) | **1.6** |

Table 3: Results on ZINC.

| Method | test MAE |
|---|---|
| GAT (Veličković et al., 2018) | .384(.007) |
| GIN (Xu et al., 2019) | .387(.015) |
| PNA (Corso et al., 2020) | .188(.004) |
| Graph Transformer (Dwivedi & Bresson, 2020) | .226(.014) |
| Transformer + RWPE (Dwivedi et al., 2021) | .310(.005) |
| SAN (Kreuzer et al., 2021) | .139(.006) |
| Graphormer (Ying et al., 2021) | .122(.006) |
| K-Subgraph SAT (Chen et al., 2022) | .094(.008) |
| Ours | .092(.001) |

Table 4: Results on OGBG-MolHIV.

| Method | AUC |
|---|---|
| GCN+VN (Kipf & Welling, 2017) | 75.99(1.19) |
| GIN+VN (Xu et al., 2019) | 77.07(1.49) |
| PNA (Corso et al., 2020) | 79.05(1.32) |
| DGN (Beaini et al., 2021) | 79.70(0.97) |
| Graphormer + FLAG (Ying et al., 2021) | 80.51(0.53) |
| SAN (Kreuzer et al., 2021) | 77.85(0.24) |
| GSN+VN (Bouritsas et al., 2022) | 77.99(1.00) |
| HMGNN + PNA (Yu & Gao, 2022) | 80.20(1.18) |
| Ours | 80.80(0.20) |

different splits used in Li et al. (2021)[1]. We also evaluate our model with the same data split used for MPG and the results can be found in the appendix.

---

[1]Note that there is a mismatch between the splits used in the original paper of MPG and the one used in the author's repository. For both cases, our model performs better than MPG under the same splits. The additional experiments are available in Appendix G.

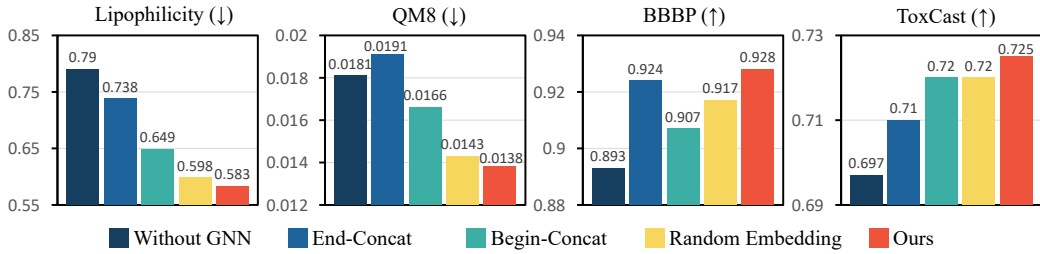

Figure 4: Ablation study. Four variations of our model (Without GNN, End-Concat, Begin-Concat, Random Embedding) on four downstream datasets. The results show the necessity of each component of our model.

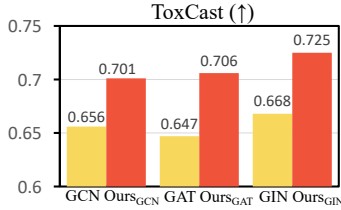

Figure 5: Comparison between standard three GNNs and our model on ToxCast dataset.

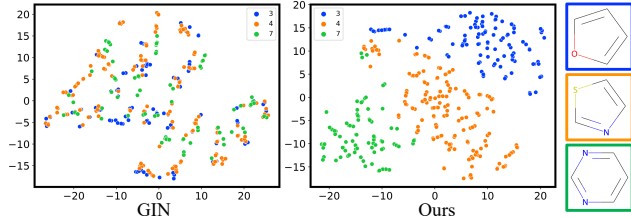

Figure 6: t-SNE embedding of molecules with different substructures.

**Results** Table 2 shows the overall results of the baselines and our model on 11 MoleculeNet datasets. Our model achieved the best performance on five downstream tasks: ClinTox, Tox21, ToxCast, ESOL and QM8, and the second best performance on five downstream tasks: SIDER, BACE, FreeSolv, Lipo and QM7. We also computed the average rank for the classification and regression tasks, separately. Our model achieved the best average rank among all compared models. The result shows that our model generalizes well across different downstream tasks, which means local information aggregated from GNN can propagate globally as interacting with structural information in Transformer. We find that some substructures weight high attention to nodes consisting of the substructures, and CLS token used for prediction also focuses on the substructures from attention scores in self-attention.

We further evaluate our model on two more molecular property prediction benchmarks, OGBG-MolHIV (Hu et al., 2020) and ZINC (Dwivedi et al., 2020). For ZINC, following the setting in Chen et al. (2022), we trained an additional slim model with a total parameter of 381,737 and we used RWPE (Dwivedi et al., 2021) added to atom features. Table 3 and Table 4 summarize the performance of our model. On both datasets, our model consistently outperforms previous Transformer and GNN based models.

### 4.3 ABLATION STUDY

We report ablation study to justify each component and flexibility of our model architecture. For the ablation study, we reduce the hyperparameter search space to only 6 learning rates {1e-3, 5e-4, 1e-4, 5e-5, 1e-5, 5e-6} to facilitate the comparison between different models. We report the test scores based on the hyperparameters from the best validation scores.

**Ablation on model components** Figure 4 shows the performance comparison between four variations of our model on four downstream datasets. We first verify the performance of our model without GNN branch. To do that, we replace all cross-attention layers with self-attention layers and exclude GNN branch. Begin-Concat and End-Concat examine different ways of combining substructure and local features. Begin-Concat runs a vanilla-transformer encoder on top of a concatenated atom and substructure embeddings. End-Concat runs a vanilla-transformer encoder on top of atom embeddings and concatenates the substructure feature at the end to make the final prediction. Random Embedding examines the effect of substructural embeddings by replacing them with random learnable embeddings. Figure 4 shows our model outperforms other variations on all datasets, which justifies the necessity of each component.

**Different GNN branch**   Our model can flexibly utilize other GNN architectures as our GNN branch. We test the changes in performance when our model design is applied with different GNN architectures. Figure 5 shows the comparison between commonly-used GNNs, i.e., Graph Convolutional Network (GCN), Graph Attention Network (GAT) (Veličković et al., 2018) and GIN, and our models with the GNN branch switched to each corresponding GNN on ToxCast dataset. There is a significant performance improvement when our model is adopted to each GNN model. Performance comparison when utilizing 3D-aware GNN is presented in Appendix F.

## 4.4   ANALYSIS

**Effectiveness of substructures**   Figure 6 shows the t-SNE embeddings of molecules with different substructures. Our model discriminates molecules with similar substructures (aromatic rings with one or two different atoms) better than GIN.

**Attention map**   Figure 7 shows the attention weights in a cross-attention and a self-attention layer. In the cross-attention layer, each row and column correspond to substructure and node in the given molecule, so we can interpret the attention score as the degree of focus between the substructures and nodes. We check where the CLS token gives more attention since it is used for prediction. We find out the CLS token gives strong attention to specific substructures. More interestingly, often a substructure gives more attention on the nodes that consists the substructure itself. Despite of not having any structural information between input substructures and nodes as input, our model can identify structurally related nodes in the cross attention layer, showcasing the ability of understanding molecular structure.

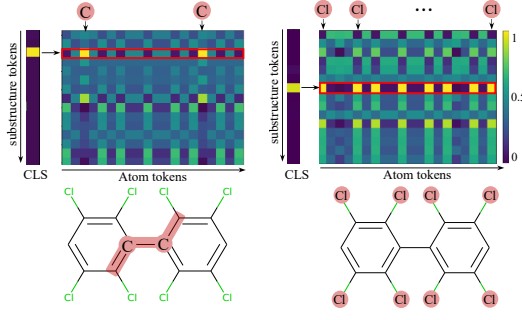

(a) Ring-Chain-Ring bond      (b) Chlorine atom

Figure 7: Attention visualization. CLS token strongly attends to Ring-Chain-Ring bond and Cl substructures of the input molecule. The cross attention maps show the substructures capture the atoms related to the substructure.

**Computation time for graph features**   Figure 8 shows the time required in millisecond to compute the MACCS keys, shortest path and 3D distance for each molecule with different number of atoms. As shown in the figure, the time required for 3D distance and the shortest path increases dramatically as the number of atoms increase, which make these features impossible to be applied in large molecules. However, identifying the MACCS key substructures does not depend on the input molecule's number of atoms.

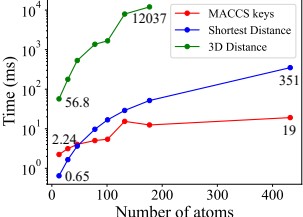

Figure 8:   Time required to compute each feature per molecule.

## 5   CONCLUSION

In this paper, we propose a novel framework that incorporates Transformer and GNN architecture for molecular representation learning. Our model takes advantages of the two architectures to aggregate substructure and local information. With the cross attention mechanism in fusing network, our model could achieve state-of-the-art performance on various molecular property prediction benchmarks. Overall, our work highlights the effectiveness of SACA for molecular representation learning. We plan to release the source code and pretrained neural networks in the public domain.

**Limitation and future work**   While we show the effectiveness of our model to predict molecular property, the proposed approach has not yet been validated with large-scale molecular graph such as proteins. Applying or modifying our model to proteins which contain 5,000 to 50,000 atoms would be an interesting direction for the future work.

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

# A   NODE AND EDGE FEATURES

In this section, we present the node and edge features of molecules used for GNN branch. We used OGB package (Hu et al., 2020) to convert SMILES strings (Weininger, 1988) to molecular graphs. The molecular graphs are encoded through GIN (Xu et al., 2019) and injected into the Transformer branch by the cross-attention. The molecular graphs have the following node and edge features.

**Node features.**   Each node has the following 9 dimensional features as shown in Table 5.

Table 5: Node Features.

| Index | Description | Range |
|-------|-------------|-------|
| 0 | Atomic num | [1, 118], other |
| 1 | Chirality | unspecified, tetrahedral cw, tetrahedral ccw, other |
| 2 | Degree | [0, 10], other |
| 3 | Formal Charge | [-5, 5], other |
| 4 | Num Hydrogen | [0, 8], other |
| 5 | Num Radical Electron | [0, 4], other |
| 6 | Hybridization | SP, SP2, SP3, SP3D, SP3D2, other |
| 7 | Is Aromatic | False, True |
| 8 | Is in Ring | False, True |

**Edge features.**   Each edge has the following 3 dimensional features as shown in Table 6.

Table 6: Edge Features.

| Index | Description | Range |
|-------|-------------|-------|
| 0 | Bond Type | single, double, triple, aromatic, other |
| 1 | Bond Stereo | stereonone, stereoz, stereoe, stereocis, stereotrans, stereoany |
| 2 | Is Conjugated | False, True |

## B  DETAILS OF DOWNSTREAM DATASETS

We used 11 binary graph classification and regression datasets: BBBP, SIDER, ClinTox, BACE, Tox21, ToxCast, FreeSolv, ESOL, Lipophilicity, QM7 and QM8 from Moleculenet (Wu et al., 2018). The details of each dataset are shown in Table 7. Through the various datasets, we can test the generalization ability of our pretrained model.

Table 7: Detailed description for each downstream dataset.

| Dataset | Description |
|---|---|
| BBBP | Binary classification task to predict a molecule's blood-brain barrier penetration ability |
| SIDER | Marketed drugs with its adverse drug reactions |
| ClinTox | Qualitative data of drugs approved by the FDA and those that have failed clinical trials for toxicity reasons |
| BACE | Binary classification task to predict a molecule's binding result for a set of inhibitors of human $\beta$-secretase 1 |
| Tox21 | Qualitative toxicity measurements on 12 biological targets |
| ToxCast | Toxicology data for a large library of compounds based on in vitro high-throughput screening, including experiments on over 600 tasks |
| FreeSolv | Regression task to predict hydration free energy of small molecules in water |
| ESOL | Regression task to predict water solubility in terms of log solubility in mols per litre |
| Lipophilicity | Experimental results of octanol/water distribution coefficient |
| QM7 | A subset of GDB-13 composed of all molecules of up to 23 atoms (including 7 heavy atoms C, N, O, and S), totalling 7165 molecules |
| QM8 | Computer-generated quantum mechanical properties |

## C  MOLECULAR STRUCTURAL KEYS

In this section, we explain the details of molecular substructures that we used for our model, i.e., Molecular ACCess System (MACCS) keys (Durant et al., 2002). We extract MACCS keys for each molecule using RDKit package (Landrum, 2016). MACCS keys have 166-dimensional features where each binary label indicates the presence of a particular substructure in the given molecule. For example, the $139^{th}$ index of MACCS keys indicates the presence of the Hydroxy group (-OH), and the $162^{nd}$ index indicates the presence of the aromatic ring in a molecule. The full list of 166 MACCS keys can be found in the document [2].

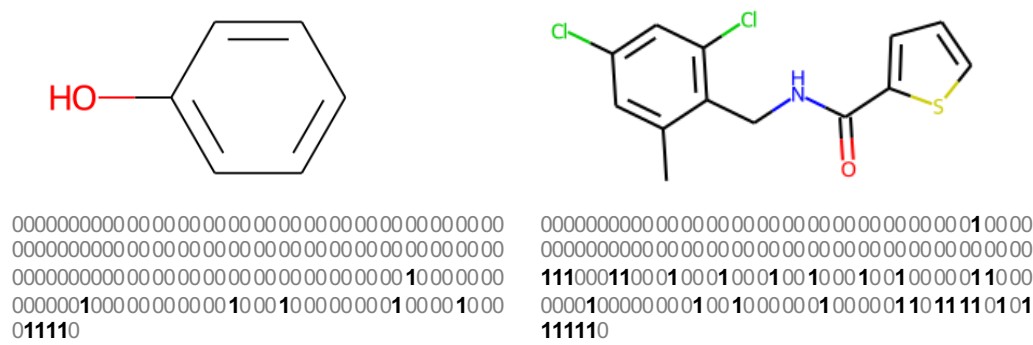

Figure 9: MACCS keys encoding.

Figure 9 shows examples of how MACCS keys encode substructure information into a binary bit string. The canonical SMILES representations of the left and right molecules are C1=CC=C(C=C1)O and Cc1cc(Cl)cc(Cl)c1CNC(=O)c1cccs1, respectively.

Table 8 shows the MACCS keys statistics for pretraining and downstream datasets. It includes the average and the median number of MACCS keys that molecules in each dataset contain. Also, the minimum and the maximum number of MACCS keys for each dataset are reported.

---

[2]https://github.com/rdkit/rdkit/blob/master/rdkit/Chem/MACCSkeys.py

Table 8: MACCS keys statistics for pretraining and downstream dataset.

| Dataset | Average | Median | Min | Max |
|---|---|---|---|---|
| Pretraining dataset | $52.12 \pm 13.33$ | 53 | 1 | 106 |
| BBBP | $46.03 \pm 14.48$ | 46 | 2 | 96 |
| SIDER | $46.61 \pm 17.93$ | 46 | 1 | 105 |
| ClinTox | $46.13 \pm 16.54$ | 46 | 2 | 92 |
| BACE | $61.04 \pm 12.53$ | 60 | 21 | 93 |
| Tox21 | $32.73 \pm 16.63$ | 30 | 2 | 99 |
| ToxCast | $33.50 \pm 17.30$ | 31 | 2 | 101 |
| FreeSolv | $15.26 \pm 9.78$ | 13 | 1 | 62 |
| ESOL | $23.63 \pm 15.52$ | 19 | 1 | 76 |
| Lipophilicity | $51.61 \pm 14.26$ | 52 | 7 | 93 |
| QM7 | $18.75 \pm 7.80$ | 18 | 1 | 50 |
| QM8 | $23.00 \pm 8.10$ | 23 | 1 | 48 |

## D EXPERIMENTAL SETTING

In this section, we explain further details of our experimental setting.

**Pretraining.** The default hyperparameters for pretraining are listed in Table 9. All attention layers have 16 attention heads and 768 hidden dimension. Our model has 41M parameters.

Table 9: Model configurations and hyperparameters for pretraining.

|  | Parameter |
|---|---|
| M | 4 |
| N | 3 |
| Model hidden dimension | 768 |
| FFN inner-layer dimension | 768 |
| # of attention heads | 16 |
| Learning rate | 0.0001 |
| Epoch | 10 |
| Dropout | 0.1 |
| Batch size | 32 |

**Hyperparameter Search Range.** For each downstream task, we search for the best hyperparameter combinations. We perform the Bayesian optimization over the validation set and use the hyperparameters for the best validation score to report the test score. The hyperparameter range that we searched over is shown in Table 10.

Table 10: Finetuning hyperparameter search range.

| Hyperparameter | Description | Range |
|---|---|---|
| Learninng rate | The learning rate | $0.000001 \sim 0.001$ |
| # of MLP layers | The number of last MLP layers | 1, 2, 3 |
| Dropout | Dropout ratio | $0.0 \sim 0.5$ |
| Weight decay | Weight decay | 0.0, 0.001, 0.0001, 0.00001 |

**Pretraining Task.** To pretrain our network, we extract the 200 real-valued descriptors for each molecule using RDKit package (Landrum, 2016). This task is proposed by (Fabian et al., 2020). Through this task, we can make the model to learn the physicochemical properties of the input molecules.

**Details to obtain pretraining dataset.** We describe the pretraining dataset collection procedure. (1) We collected 2,271,376 unlabeled molecules from ChEMBL and PubChem databases. (2) We canonicalized the SMILES using RDKit to obtain unique representation of molecules and remove all duplicated molecules in the pretraining dataset. (3) We retrieved all overlapping molecules between the pretraining and downstream datasets. (4) We removed the overlapping molecules from the pretraining dataset and keep the downstream dataset intact. Such a procedure results in 2,271,376 - 413,295 = 1,858,081 molecules in the pretraining dataset. Table 11 shows the statistics of overlapping molecules between each downstream dataset and the originally collected pretraining dataset.

Table 11: Overlapping molecules between original pretraining dataset and each downstream dataset.

|                    | BBBP  | SIDER | Clintox | BACE | Tox21 | ToxCast | FreeSolv | ESOL  | Lipo  | QM7  | QM8   |
|--------------------|-------|-------|---------|------|-------|---------|----------|-------|-------|------|-------|
| # mols             | 2039  | 1427  | 1478    | 1513 | 7831  | 8576    | 642      | 1128  | 4200  | 6830 | 21786 |
| Overlapping # mols | 1029  | 956   | 631     | 3    | 7105  | 8243    | 459      | 826   | 1260  | 529  | 869   |
| Ratio              | 50.5% | 67.0% | 42.7%   | 0.2% | 90.7% | 96.1%   | 71.5%    | 73.2% | 30.0% | 7.7% | 4.0%  |

## E  PARAMETERS AND SPACE COMPLEXITY OF DIFFERENT MODELS

We present the number of parameters of transformer-based molecular representation learning models in Table 12. Except for CoMPT, our model has a comparable or lower number of parameters than other transformer-based models.

We also present the space complexity in Table 12. The space complexity for our attention computation is linear to the number of atoms (i.e., $\mathcal{O}(n)$ where $n$ is the number of atoms). This is because the number of MACCS key, the substructure that is used in our model, is bounded by 166. On the other hand, the space complexity of other transformer-based molecule representation learning models is quadratic to the number of atoms (i.e., $\mathcal{O}(n^2)$).

Table 12: Parameter and complexity comparison between different models.

|                              | Parameters | Space Complexity for Attention |
|------------------------------|------------|--------------------------------|
| GROVER (Rong et al., 2020)   | 48M        | $\mathcal{O}(n^2)$             |
| Graphormer (Ying et al., 2021)| 47M       | $\mathcal{O}(n^2)$             |
| MAT (Maziarka et al., 2020)  | 42M        | $\mathcal{O}(n^2)$             |
| CoMPT (Chen et al., 2021)    | 2.7M       | $\mathcal{O}(n^2)$             |
| Ours                         | 41M        | $\mathcal{O}(n)$               |

## F  ADDITIONAL ABLATION STUDY

**Change GNN branch to 3D-aware GNN** To observe the effect of the choice of GNN architecture for GNN branch, we conduct additional experiments with 3D-aware GNN. We changed our GNN branch from GIN to SGCN (Danel et al., 2020) that utilizes 3d coordinates of input molecules. Please note that the two models are not pre-trained and instead trained on the downstream dataset from scratch. Figure 10 shows that the model with 3D-aware GNN shows better performance than our original model on QM7 dataset whose task is closely related to 3D information. The result shows the flexibility of our framework that can be further tuned by using task-appropriate GNN architecture.

**Other substructures as input tokens for Transformer** Despite of the fact that our model utilizes MACCS keys for the input of Transformer, our model can flexibly receive any substructure vocabulary. We conduct experiments using other substructures from ECFP fingerprints, Tree decomposition and Pubchem Fingerprint. Table 13 summarize the results. While other substructures achieve competitive performance on most of the downstream tasks, MACCS keys shows the best performance on average.

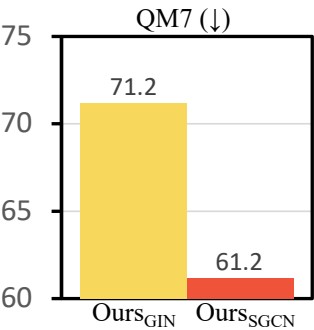

Figure 10: Comparison of our model where the GNN branch is replaced with 3D GNN.

We presumed the performance decrease when using ECFP or Tree decomposition as follows. First, they do not encode explicit pre-defined functional groups as their motifs. Particularly, ECFP defines motifs as a subgraph within a certain diameter around each node. This mechanism is similar to how GNN encodes node representations as they both see the local subgraph information around each node. As we utilize GNN branch, as a separate branch, the use of ECFP fingerprint might not bring new information. Secondly, the possible number of substructures for data-driven motifs are much larger than MACCS key. Specifically, ECFP commonly encodes molecules into 1024 bits, which might make it challenging for the model to learn the relationship between each substructure and node. Also, Pubchem Fingerprint defines a larger substructure vocabulary of 881 than MACCS key. The result shows that larger vocabulary does not always bring performance improvement.

Table 13: Comparison between different substructures applied to our model.

| Method | BBBP ↑ | BACE ↑ | Tox21 ↑ | ToxCast ↑ | FreeSolv ↓ | ESOL ↓ | Lipo ↓ | QM8 ↓ |
|---|---|---|---|---|---|---|---|---|
| MACCS key | 0.934 | 0.868 | 0.818 | 0.725 | 2.00 | 0.878 | 0.582 | 0.0140 |
| ECFP 4 | 0.925 | 0.869 | 0.818 | 0.716 | 2.52 | 0.900 | 0.596 | 0.0152 |
| ECFP 6 | 0.903 | 0.861 | 0.818 | 0.709 | 2.30 | 0.949 | 0.592 | 0.0151 |
| Tree Decomposition | 0.925 | 0.848 | 0.796 | 0.715 | 2.37 | 0.885 | 0.614 | - |
| Pubchem Fingerprint | 0.922 | 0.853 | 0.802 | 0.709 | 1.83 | 0.977 | 0.719 | - |

## G PERFORMANCE ON MPG SPLIT

As MPG uses different splits to GROVER on the paper, we reproduce the result of our model on the same split that MPG used for fair comparison. As shown in Table 14, our model outperforms MPG on most of the downstream datasets.

Table 14: Comparison between MPG and our model on the split from MPG paper.

| Method | BBBP ↑ | SIDER ↑ | Clintox ↑ | BACE ↑ | Tox21 ↑ | ToxCast ↑ | FreeSolv ↓ | ESOL ↓ | Lipo ↓ |
|---|---|---|---|---|---|---|---|---|---|
| MPG | 0.922 | 0.661 | 0.963 | 0.920 | 0.837 | 0.748 | 1.269 | 0.741 | 0.556 |
| Ours | 0.943 | 0.665 | 0.958 | 0.931 | 0.847 | 0.748 | 1.278 | 0.719 | 0.546 |

## H MODEL VARIANTS

In this section, we illustrate the difference between 4 variants of models that utilize molecular substructures, as shown in Figure 11. The first model is Extra Atom-level feature, which concatenates substructure features to atom features that make up the substructure, and forwards them through the Transformer. The second model is Begin-Concat, which tokenizes substructures along with atom tokens and uses them as input for the model. The third model is End-Concat, and this model forwards only atom tokens through Transformer, followed by concatenating substructure feature before the last fully-connected layer. The fourth model is our proposed model, which forwards substructure tokens through Transformer, node representations through a separate GNN and mixes these through cross attention.

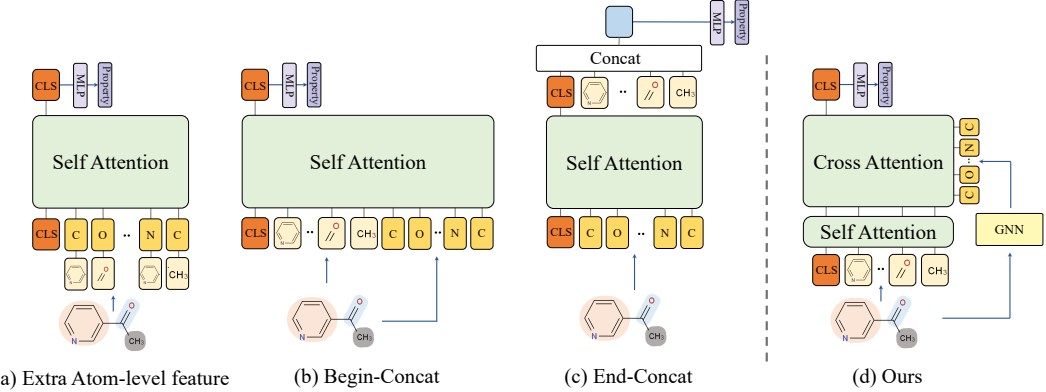

Figure 11: Model variants that incorporate molecular substructures and atoms.

These different ways of mixing substructures and atoms have been explored in existing work (Fang et al., 2021; Cai et al., 2022; Wu et al., 2021a) and the first three variants have some limitations. First, Extra Atom-level feature makes it difficult for the model to separate the semantic meanings of substructures from atoms only. For example, the Hydroxy group (-OH) is important by itself, but it is hard to obtain this substructure information from hydrogen and oxygen atom features implicitly. Second, in Begin-Concat, the direct concatenation of substructure and atom tokens has the limitation that the model does not differentiate the two different types of tokens. A full self attention is computed between concatenated substructure and atom tokens at the same level. Lastly, in End-Concat, there is no interaction between substructures and atoms except for the last linear layer. In subsection 4.3, we have also shown Begin-Concat and End-Concat have poor performance than our model, which fuses two types of information through cross attention. This exploration of model variants shows our model proposes a good way to mix the two types of information.

# I OTHERS

**URLs for pretrained model for baselines.**  We use the following sources for pretrained models to reproduce the results.

Graphormer: `https://github.com/microsoft/Graphormer`

MAT: `https://github.com/ardigen/MAT`

GraphLoG: `https://github.com/DeepGraphLearning/GraphLoG`

GEM:  `https://github.com/PaddlePaddle/PaddleHelix/tree/dev/apps/pretrained_compound/ChemRL/GEM`

