# OpenReview forum: "Substructure-Atom Cross Attention for Molecular Representation Learning"
_ICLR.cc/2023/Conference — Submitted to ICLR 2023_

### Official Review · Reviewer_bDKw · 2022-10-23

**Confidence:** 3
**Correctness:** 4
**Technical Novelty And Significance:** 2
**Empirical Novelty And Significance:** Not applicable
**Recommendation:** 5

**Clarity, Quality, Novelty And Reproducibility:**

The novelty of the work is detailed in the "strength and weakness" section. The paper is written fairly clearly. However, I disagree that the time complexity of the cross-attention is O(N). Based on equation 1, the cross attention does not include the substructure-atom membership information. Each substructure looks at all atoms rather than atoms that belong to its substructure. Therefore, if there are N substructures and M atoms, the cross attention should be O(NM). I would recommend the authors to clarify on this point.


**Strength And Weaknesses:**

Strength
* The empirical evaluation is very comprehensive and includes most of the state-of-the-art baselines.
* The results showed that the method outperforms all the baselines (averaged over 15 datasets).

Weakness
* The novelty of the proposed architecture is a bit limited. Similar architectures have been proposed before, e.g. MolFormer [1]. It would be good to discuss the technical differences and compare with MolFormer on the same datasets.
* The proposed architecture treats each substructure as a discrete token and there is no relational attention between two substructures (e.g., the distance between two substructures, how many shared atoms, etc). In other words, the substructure transformer is a set transformer with no relational information between the substructures. It that's the case, the proposed model is essentially a transformer over MACCS keys + GNN + cross attention between the two. I believe that relational attention should be included, which could further improve the method.

[1] Molformer: Motif-based Transformer on 3D Heterogeneous Molecular Graphs, arxiv 2021

**Summary Of The Paper:**

This paper proposes a transformer-based architecture for molecular property prediction. The proposed method represents a molecule at two different levels: substructure-level and atom-level. The substructures of a molecule are extracted through MACCS fingerprint. Each substructure is embedded as a token and encoded by a transformer. The atom-level embedding is learned through a standard GNN and it is fused into the substructure embedding. The model pre-trained on a large collection of unlabelled molecules and fine-tuned on MoleculeNet benchmark datasets.

**Summary Of The Review:**

Overall, there are several concerns of the paper that needs to be addressed:
1) Novelty compared to MolFormer and other previous work
2) The cross attention is not linear time complexity based on equation 1, which contradicts with author's claim.
Therefore, I vote for weak rejection of this paper.

---

> ### Author Response · Authors · 2022-11-23
> **Response to Reviewer bDKw**
>
> We deeply appreciate your insightful comments and constructive feedback. Here, we address your concerns as follows:
>
>
> **1. The novelty of the proposed architecture is a bit limited. Similar architectures have been proposed before, e.g. MolFormer [1]. It would be good to discuss the technical differences and compare with MolFormer on the same datasets.**
>
> Thank you for suggesting a good baseline to compare. We think there is a key difference between our model and MolFormer [1]. Please note that we illustrated the difference between MolFormer and our model in section H of the Appendix (i.e., Begin-Concat vs. Ours). We also reproduced Molformer on our setting and showed that our model outperforms Molformer, as shown in Table 2 of the revised paper.
>
> MolFormer uses motif embeddings along with atoms as the input of the Transformer, but embeds motifs as a parallel line of atoms tokens and forwards them through a transformer all at once. This is similar to one variant of our model named as Begin-Concat where we embed atoms and motifs and forward them through a vanilla transformer. It is presented in Figure 4 in Section 4.3. In our ablation study, the direct concatenation of motifs and atoms, either in the beginning or the end, shows sub-optimal performance compared to our model, which fuses two different types of information through cross attention. By computing attention between substructures and atoms, our model can understand the relation between semantic motifs (substructures) and atoms and aggregate structurally important node features.
>
> **2. The proposed architecture treats each substructure as a discrete token and there is no relational attention between two substructures (e.g., the distance between two substructures, how many shared atoms, etc ). In other words, the substructure transformer is a set transformer with no relational information between the substructures. It that's the case, the proposed model is essentially a transformer over MACCS keys + GNN + cross attention between the two. I believe that relational attention should be included, which could further improve the method.**
>
> Thank you for the constructive suggestion. In our work, we mainly focused on the effect of combining substructures and node features. Nevertheless, we tried to incorporate positional information for substructures, such as the distance between them, but it was not straightforward to define the distances of substructures. However, as suggested, it would be a good approach to use the number of shared atoms between two substructures in self attention for relational information. We leave this for future work to further develop our model.
>
> **3. However, I disagree that the time complexity of the cross-attention is O(N). Based on equation 1, the cross attention does not include the substructure-atom membership information. Each substructure looks at all atoms rather than atoms that belong to its substructure. Therefore, if there are N substructures and M atoms, the cross attention should be O(NM). I would recommend the authors to clarify on this point.**
>
> The model complexity in the cross-attention map depends on how substructures are extracted. Since the maximum number of substructures for MACCS key is 166, we denoted O(n) ~ O(166n), which is one reason for using the MACCS key to reduce model complexity. If we use data-driven substructures (e.g., BRICS fragmentation) whose vocabulary size increases linearly as the number of atoms increases, then the computational complexity becomes O($n^2$).
>
> Reference
>
> [1] Fang, Dragomir, et al. "MolFormer: Motif-based Transformer on 3D Heterogeneous Molecular Graphs." arXiv preprint arXiv:2110.01191

---

### Official Review · Reviewer_JtFD · 2022-10-24

**Confidence:** 5
**Correctness:** 3
**Technical Novelty And Significance:** 2
**Empirical Novelty And Significance:** 2
**Recommendation:** 5

**Clarity, Quality, Novelty And Reproducibility:**

The paper was well written. Novelty and technical contribution are limited. The code was not included in the submission.

**Strength And Weaknesses:**

Strengths
The paper is well written. The results look promising. The idea of the fusion of the representation from two different architectures to leverage the strengths of both methods is a good practice worth sharing with the application community.

The examples in Figure 1 are an interesting motivation example for leveraging motif substructure in molecules to improve the disadvantage of GAT GNN.

Weaknesses
This is rather a simple approach trivially combining existing techniques. For the ICLR conference, I would like to see more depth on technical contribution. I would suggest comparing the following simple baselines:

 + Ensemble of GNN and Transformers Models, just consider a linear model with the same weight as all the baseline models

+ Recent works have considered learning a space that combines information from 2D and 3D data GraphMVP, please look at https://arxiv.org/abs/2110.07728, this is a good baseline as GraphMVP help fix the issues that models learned from 2D data that miss information about the 3D arrangement of the atoms in the molecules.


Potential leakage during pretraining. It is important to remove test molecules from the data used for pretraining. This removal is important to make sure that there is no leakage happens.



**Summary Of The Paper:**

Summary of the paper
The paper proposed a simple molecule representation learning method that combines representation from a transformer and a GNN network to leverage the strengths of each method. While transformers are good at learning frequent substructures or motifs in a molecule database, GNN provides additional information about missing structures due to the linearization of molecule graphs.

The authors also pretrained their proposed networks with 1.8 M molecules with the objective of predicting 200 real-valued descriptors of physical-chemical properties from the pretraining datasets using RDKit.

The experiments on 11 downstream tasks demonstrated significant results  of the proposed methods compared to different transformers  and GNN networks both with pretrained/non-pretrained models







**Summary Of The Review:**

Summary of my comments
The paper was well-written and I see that its method for combining strengths from two different representations of learning architecture is an interesting practice but a combination of these two known techniques is not technically strong enough for an ICLR paper.
I would recommend the authors validate the fusion approach against a simple linear ensemble of the baseline methods listed in your paper. Also, it is important to remove the test molecules of the benchmark datasets from the pretraining data.

---

> ### Author Response · Authors · 2022-11-23
> **Response to Reviewer JtFD**
>
> We deeply appreciate your efforts and insightful comments. Here, we address the comments and questions mentioned.
>
> **1. This is a rather simple approach, trivially combining existing techniques. For the ICLR conference, I would like to see more depth on technical contributions. I would suggest comparing the following simple baselines:**
> - **Ensemble of GNN and Transformers Models, just consider a linear model with the same weight as all the baseline models.**
> - **Recent works have considered learning a space that combines information from 2D and 3D data GraphMVP, please look at https://arxiv.org/abs/2110.07728, this is a good baseline as GraphMVP help fix the issues that models learned from 2D data that miss information about the 3D arrangement of the atoms in the molecules.**
>
> Thank you for suggesting additional baselines to compare. We further report the performance of the ensemble model with GIN and GROVER on several downstream datasets. We implement two methods of the ensemble: 1) soft voting with prediction values from GIN and GROVER, and 2) adding MLP head on top of two models after removing the last prediction layers. Table 1 shows the results. The result consistently shows that ensemble methods do not lead to performance improvement but rather follow a model which has a better performance. On the other hand, our model, which fuses two types of information from Transformer and GNN through cross attention, achieves good performance across all datasets. Also, we have validated the effectiveness of our model compared to naive baseline models (i.e., Begin-Concat and End-Concat). Please check Section H in the Appendix of the revised paper.
>
> We also reproduced the mentioned baseline GraphMVP on the MoleculeNet dataset, and the result shows that our model outperforms GraphMVP as shown in Table 2 of Section 4.2.
>
>
> Table 1. Performance on ensemble models.
>
> |   Dataset   |  BBBP (↑) |  BACE (↑) | ClinTox (↑) |
> |:-----------:|:---------:|:---------:|:-----------:|
> |     GIN     |   0.904   |   0.836   |    0.855    |
> |    GROVER   | **0.936** | **0.878** |  **0.925**  |
> | Soft Voting |   0.931   |   0.875   |    0.919    |
> |     MLP     |   0.895   |   0.776   |    0.865    |
>
> **2. Potential leakage during pretraining. It is important to remove test molecules from the data used for pretraining. This removal is important to make sure that there is no leakage happens.**
>
> We made sure that there were no overlapping molecules between pretraining and downstream datasets before the experiment. Therefore, we believe there is no data leakage from pretraining for each downstream task. We describe the pretraining dataset collection procedure. (1) We collected 2,271,376 unlabeled molecules from ChEMBL and PubChem databases. (2) We canonicalized the SMILES using RDKit to obtain a unique representation of molecules and remove all duplicated molecules in the pretraining dataset. (3) We retrieved all overlapping molecules between the pretraining and downstream datasets. (4) We removed the overlapping molecules from the pretraining dataset and kept the downstream dataset intact. The procedure results in 2,271,376 - 413,295 = 1,858,081 molecules in the pretraining dataset. Table 2 shows the statistics of overlapping molecules between each downstream dataset and the originally collected pretraining dataset. We add the details of the preprocessing steps in Section D of the Appendix.
>
> Table 2. Overlapping molecules between the original pretraining dataset and each downstream dataset.
>
> |                    | BBBP  | BACE | ToxCast | Tox21 | ClinTox | SIDER | FreeSolv | ESOL  | QM8   | QM7  | Lipo  |
> |--------------------|-------|------|---------|-------|---------|-------|----------|-------|-------|------|-------|
> | # mols             | 2039  | 1513 | 8576    | 7831  | 1478    | 1427  | 642      | 1128  | 21786 | 6830 | 4200  |
> | Overlapping # mols | 1029  | 3    | 8243    | 7105  | 631     | 956   | 459      | 826   | 869   | 529  | 1260  |
> | Ratio              | 50.5% | 0.2% | 96.1%   | 90.7% | 42.7%   | 67.0% | 71.5%    | 73.2% | 4.0%  | 7.7% | 30.0% |

---

### Official Review · Reviewer_cHkX · 2022-10-24

**Confidence:** 3
**Correctness:** 4
**Technical Novelty And Significance:** 3
**Empirical Novelty And Significance:** Not applicable
**Recommendation:** 8

**Clarity, Quality, Novelty And Reproducibility:**

# Clarity, Quality, Novelty, and Reproducibility
## Clarity
The paper is clear and well written.

## Quality
The experiments used appear to be of high quality and well support the final conclusions.

## Novelty
To the best of my knowledge, this paper is meaningfully novel, and the presented final results certainly suggest this approach has significant merit.

## Reproducibility
This may be challenging to reproduce given the complexity of the problem, and it is not stated in the work whether or not code is planned to be released.


**Strength And Weaknesses:**

# Strengths and Weaknesses

## Key Strengths (reasons I would advocate this paper be accepted)
  1. I think this approach is well motivated, aimed towards an impactful problem, and well presented.
  2. The results you present are compelling, and compare against meaningful baselines using appropriate evaluation metrics.

## Key Weaknesses (reasons I would advocate this paper be rejected)
  None.

## Minor Strengths (things I like, but wouldn't sway me on their own)
  1. I like that you include ablation studies motivating the design choices of your system.

## Minor Weaknesses (things I dislike, but wouldn't sway me on their own)
  1. There are some missing details; for example, how are positional encodings generalized for their substructure transformer in this case? In addition, while hyperparameter search is described, architecture search is not (e.g., how many layers are used).


**Summary Of The Paper:**

# Summary
Substructure-Atom Cross Attention for Molecular Representation Learning

## What is the problem?
Prediction of molecular properties via deep learning methods is important for AI-aided drug discovery.

## Why is it impactful?
Drug discovery is expensive, so better machine learning methods to solve this task could help accelerate drug discovery at lower costs.

## Why is it technically challenging/interesting (e.g., why do naive approaches not work)?
Processing molecular graphs is challenging, as important features exist at various scales within the input graphs, and we have a puacity of labeled data to train high-capacity methods.

## What is this paper's contribution?
This paper proposes to use extracted molecular substructures (extracted via MACCS keys) and atom-level features simultaneously via a two-branched neural network approach, where molecular substructures are processed via a transformer architecture and atoms via a GNN architecture.

In addition, the authors use a large-scale multi-task pre-training approach to initialize their architecture, thereby helping ameliorate for the small dataset scale available here.

## How do these methods compare to prior works?
While there are prior works that use both GNNs and transformers together, and approaches that leverage graph transformers directly, I am not aware of any papers that jointly leverage atomic features and MACCS substructures simultanesouly.

## How do they validate their contributions?
The authors validate their contribution against a variety of baselines from the MoleculeNet benchmark, finding that their approach achieves rank 1.8 on average and 1.6 on average in classification and regression tasks, respectively. Baselines appear to be compelling, and span both pre-trained and non-pre-trained approaches.


**Summary Of The Review:**

I think this is a very strong, well-written paper proposing a meaningful solution to an important problem and demonstrating the utility of that solution via compelling results.

---

> ### Author Response · Authors · 2022-11-23
> **Response to Reviewer cHkX**
>
> We deeply appreciate your efforts and positive comments. Here, we address the comments and questions mentioned.
>
> **1. There are some missing details; for example, how are positional encodings generalized for their substructure transformer in this case? In addition, while hyperparameter search is described, architecture search is not (e.g., how many layers are used).**
>
> Thank you for the question. Please note that **we did not use any positional encoding for the substructures in Transformer**. In fact, it is not trivial to define the positions of substructures. Instead, our GNN branch takes the role of the positional encodings by injecting node features, including k-hop local information, into the Transformer. With this model architecture, we can avoid expensive computational overhead to obtain graph features. This aspect is the key benefit of our approach, as demonstrated in Figure 8 in Section 4.4. Nevertheless, adding positional encodings and considering the relation between substructures would be an interesting research direction. We leave this for future work.
> For architectural design, we mostly followed the model configurations (e.g., number of hidden dimensions) of one representative molecular transformer architecture, Graphormer [1]. For the GNN model configuration, we followed Hu et al [2].
>
>
> **2. This may be challenging to reproduce given the complexity of the problem, and it is not stated in the work whether or not code is planned to be released.**
>
> As we mentioned in the last bullet point of the introduction, source code and pretrained networks will be released in the public domain upon the paper's acceptance.
>
>
> Reference
>
> [1] Ying et al., Do Transformers Really Perform Bad for Graph Representation? (NeurIPS 2021)
>
> [2] Hu et al., Strategies for Pre-training Graph Neural Networks (ICLR 2020)

---

### Official Review · Reviewer_vSFi · 2022-10-25

**Confidence:** 4
**Clarity, Quality, Novelty And Reproducibility:** Due to the existing related works, th…
**Correctness:** 3
**Technical Novelty And Significance:** 2
**Empirical Novelty And Significance:** 2
**Recommendation:** 5

**Strength And Weaknesses:**

Pros:
1. The paper is well-organized and easy to follow.
2. The ablation study is well-designed.

Cons:
1. The idea of combining the information from molecular fingerprints and node embeddings learned from GNN is not novel and has been proposed in [1, 2].
2. The results shown in Table 2 are not convincing enough. The authors use random scaffold split and rerun the experiments 3 times following GROVER. Since the results between GROVER and the proposed model are very close (usually smaller than the standard deviation), it would be necessary to know if the differences are statistically significant. Otherwise, a fixed scaffold split may be used for fair comparisons. Besides, when comparing the proposed model to MPG, the proposed model outperforms MPG as shown in Table 2, while their performances are on par as shown in Table 2. I'm concerned about whether the other models would also have such different performances when changing the splitting way. Lastly, there is work [3] using pretraining strategy that has better results on the datasets being compared.
3. The authors claim that their model has O(n) complexity according to the self and cross attention map in the Transformer branch. However, for the cross attention map, the complexity is actually O(nm), where n is the number of atoms, and m is the number of substructures. Since the number of substructures should be linear to the number of atoms in a molecule, I think the resulting complexity should be O(n^2).
4. There is a mismatch between the shape of the rectangles and the size of the matrics as shown in Figure 3. For example, a 3*4 grid and a 2*4 grid all represent an m*d matrix, which is misleading.

[1] Fang, Y., Yang, H., Zhuang, X., Shao, X., Fan, X. and Chen, H., 2021. Knowledge-aware contrastive molecular graph learning. arXiv preprint arXiv:2103.13047.

[2] Cai, H., Zhang, H., Zhao, D., Wu, J. and Wang, L., 2022. FP-GNN: a versatile deep learning architecture for enhanced molecular property prediction. arXiv preprint arXiv:2205.03834.

[3] Xia, J., Zheng, J., Tan, C., Wang, G. and Li, S.Z., 2022. Towards effective and generalizable fine-tuning for pre-trained molecular graph models. bioRxiv.

**Summary Of The Paper:**

This paper proposes a framework that incorporates Transformer and GNN for molecular representation learning. In particular, the Transformer branch encodes molecular substructures by using molecular fingerprints as input, and the GNN branch extracts local node features from molecular graphs to provide embeddings for the fusion network's cross-attention in the Transformer branch.

**Summary Of The Review:**

Given the limited novelty and the unconvincing comparisons to the baselines, I would not recommend an acceptance.

---

> ### Author Response · Authors · 2022-11-23
> **Response to Reviewer vSFi (1/2)**
>
> We deeply appreciate your efforts and insightful comments, including editorial comments to improve the manuscript. Here, we address the comments and questions mentioned.
>
> **1. The idea of combining the information from molecular fingerprints and node embeddings learned from GNN is not novel and has been proposed in [1, 2].**
>
> Thank you for suggesting additional baselines to validate the contributions of our model. [1] incorporates substructure information into atom features, whereas [2] encodes molecular fingerprints through a separate linear layer and concatenates these features to the output of the GNN network. The difference between models is illustrated in Figure 11 in Appendix H ((a) Extra Atom-level feature for [1], (c) End-Concat for [2]). We believe there is a significant difference between the proposed baselines and our model. Our model encodes substructures through self attention and node representations through a separate GNN and then incorporates these two types of information through cross attention. As a result, our model can understand semantic motifs (substructures) and their relation with atoms and aggregate structurally important nodes.
>
> In addition, the two baselines [1], [2] have some limitations. First, [1] makes it difficult for the model to separate the semantic meanings of substructures from atoms only. For example, the Hydroxy group (-OH) is important by itself, but it is hard to obtain this substructure information from hydrogen and oxygen atom features implicitly. Secondly, [2] encodes substructures and molecular graphs separately. Hence there is no interaction between these two types of information except for the last linear layer. We have shown that End-Concat, which is similar to [2], shows worse performance than our model in Figure 4 in Section 4.3.
>
> **2. The results shown in Table 2 are not convincing enough. The authors use random scaffold split and rerun the experiments 3 times following GROVER. Since the results between GROVER and the proposed model are very close (usually smaller than the standard deviation), it would be necessary to know if the differences are statistically significant. Otherwise, a fixed scaffold split may be used for fair comparisons.
> Besides, when comparing the proposed model to MPG, the proposed model outperforms MPG as shown in Table 2, while their performances are on par as shown in Table 2. I'm concerned about whether the other models would also have such different performances when changing the splitting way.**
>
> Thank you for the opportunity to clarify this point. We clarify that we use three random scaffold splits for rerunning the experiments, but **we fix the three split seeds, the same seeds that GROVER utilized, across the evaluation of different models**. Therefore, there is **no unnecessary randomness** introduced from the dataset splitting procedure. We also note that our three splits help to alleviate the reviewer’s concern about the model performance being different when changing the splitting way, i.e., evaluation being biased to a single dataset split.
>
> In addition, we want to point out that although the performance between GROVER and our model is close, it is nontrivial to consistently achieve good performance in various classification and regression tasks, as the average ranking of our approach indicates. To further resolve the concern, we evaluated our model on two more widely-used molecular property prediction datasets (i.e., OGBG-MolHIV and ZINC). The results are shown in Table 3 and 4 in Section 4.2. Our model achieves the best performance over all baselines.
>
> **3. Lastly, there is work [3] using pretraining strategy that has better results on the datasets being compared.**
>
> Thanks for introducing a paper, but we believe this paper is not directly comparable to our paper since the paper suggests **finetuning strategies**, rather than **pretraining strategies**. The mentioned work [3] proposes two finetuning strategies on top of the pre-trained network MPG [4]; 1) MolAug to enrich the downstream molecular datasets by augmentation, 2) WordReg to improve the regularization ability of the model. Therefore, we believe [3] is not a directly comparable model to our work. As it applies these strategies on top of the pre-trained network MPG, we need to apply these strategies to our model as well for a fair comparison, but it is not possible to reproduce claimed performance enhancement due to the inability of the code. We also would like to point out that our model outperformed MPG [4] model that [3] used as a base model.

---

> > ### Author Response · Authors · 2022-11-23
> > **Response to Reviewer vSFi (2/2)**
> >
> > **4. The authors claim that their model has O(n) complexity according to the self and cross attention map in the Transformer branch. However, for the cross attention map, the complexity is actually O(nm), where n is the number of atoms, and m is the number of substructures. Since the number of substructures should be linear to the number of atoms in a molecule, I think the resulting complexity should be O(n^2).**
> >
> > The model complexity in the cross-attention map depends on how substructures are extracted. Since the maximum number of substructures for MACCS keys is 166, we denoted O(n) ~ O(166n), which is one reason for using the MACCS key to reduce model complexity. If we use data-driven substructures (e.g., BRICS fragmentation) whose vocabulary size increases linearly as the number of atoms increases, then the computational complexity becomes O(n^2).
> >
> > **5. There is a mismatch between the shape of the rectangles and the size of the matrics, as shown in Figure 3. For example, a 34 grid and a 24 grid all represent an m\*d matrix, which is misleading.**
> >
> > Thank you for letting us know about the error in Figure 3. We have fixed it in the revised version.
> >
> >
> > Reference
> >
> > [1] Fang, Yin, et al. "Knowledge-aware contrastive molecular graph learning." arXiv preprint arXiv:2103.13047
> >
> > [2] Cai, Hanxuan, et al. "FP-GNN: a versatile deep learning architecture for enhanced molecular property prediction." arXiv preprint arXiv:2205.03834
> >
> > [3] Xia, Jun, et al. "Towards effective and generalizable fine-tuning for pre-trained molecular graph models." bioRxiv
> >
> > [4] Li, Pengyong, et al. "An effective self-supervised framework for learning expressive molecular global representations to drug discovery." Briefings in Bioinformatics 22.6 (2021)

---

### Official Review · Reviewer_Jvvo · 2022-10-25

**Confidence:** 5
**Correctness:** 4
**Technical Novelty And Significance:** 4
**Empirical Novelty And Significance:** 3
**Recommendation:** 6

**Clarity, Quality, Novelty And Reproducibility:**

This paper's writing is very good and carefully organized. The figures and examples are clean and persuasive. They promised in the Introduction that "The source code and pretrained networks will be released in the public domain upon the paper acceptance", and reproducibility would follow.

As for the novelty, although individual techniques are widely known, the paper presents an interesting, simple, and novel architecture by effectively combining relevant techniques.


**Strength And Weaknesses:**

[Strength]

- The paper presents a simple, easy-to-understand, and effective design pattern to fuse Transformers and GNNs for molecular tasks with good empirical supports. It can incorporate the predefined knowledge of motif vocabulary into representation learning from molecules.

- It was very nice to see the presented method didn't try to explicitly match the predefined subgraph motifs to the input molecules, and this enables a computationally efficient and clean architecture. It would require large-scale pretraining but the relationship between motifs and input molecules is indirectly acquired in a data-driven way. In contrast, typical research considering substructure-aware methods often requires a bit costly computations such as graph matching, subgraph isomorphism, motif tree compositions, etc.

- The details are carefully followed by ablation studies (without GNN vs End-Concat vs Begin-Concat vs Random Embedding vs proposed, the results when we use different GNNs instead of GIN), experimental validations cover a wider range of competitive methods including several Graph Transformers, the pretraining is done with a large 1.8M record dataset collected from ChEMBL and PubChem, complementary analysis for practical uses such as visualization of learned features by tSNE, visualization of attention weights, and computation time.

[Weaknesses]

- One of the concerns is a potential risk of data leakage in evaluations. The paper's empirical studies are grounded on the test performance on 11 tasks of MoleculeNet. But the base model is pretrained with a large (self-collected?) dataset of 1,858,081 molecules from PubChem and ChEMBL that might also cover some/large parts of the MoleculeNet dataset. Though I understand that the pretext tasks for pretraining (to predict 200 RDKit-calculated molecular descriptors) are completely different from the MoleculeNet tasks.

- Another concern is about the use of MACCS keys. It is a set of 166 keys and usually considered to be a bit weak to represent molecules in the entire ChEMBL and PubChem database. This is basically why PubChem developed their own version of substructural keys called PubChem Fingerprint that is a 881-bit-long structural key defined at https://pubchemdocs.ncbi.nlm.nih.gov/data-specification

If the input molecule doesn't have any of 166 structural keys, the input to Transformer branch (the main branch) becomes empty, and this model doesn't make any sense. This would be one of the most well-known disadvantages to use predefined motifs (as in MACCS keys) rather than motifs occurring in the dataset (as in ECFP). So what was the average number of on-bit keys out of 166 motifs? This case would not be likely to happen?

Also, it seems no problem if we just use data-driven motifs, by BRICS or ECFP, if the on bits are hashed into a fixed number of bits. Have you tested any other options than MACCS keys, particularly data-driven ones such as BRICS motifs or ECFP keys?

- Related to the above point, the following three related methods [1] [2] [3], which were not included in the empirical evaluation of the paper ([1] was cited though), use data-driven motifs with GNNs or Transformers or both. Of particular interest, the "Structure-Aware Transformer (SAT)" [3] is based on Transformer + GNN like, First, extract k-hop subgraphs, process them by GNNs, and fed outputs into Transformer as Q and K with V as input molecule. So it would be highly appreciated to discuss the relationship. Is this pattern "End-Concat"..? (First, motifs occurring in the input molecule are fed into GNNs, and their outputs are fed into Transformer. See the picture at https://github.com/BorgwardtLab/SAT)

[1] Zhang et al, Motif-based Graph Self-Supervised Learning for Molecular Property Prediction. (NeurIPS 2021)
    https://arxiv.org/abs/2110.00987

[2] Yu and Gao, Molecular Representation Learning via Heterogeneous Motif Graph Neural Networks (ICML2022)
    https://arxiv.org/abs/2202.00529

[3] Chen et. al, Structure-Aware Transformer for Graph Representation Learning (ICML2022)
    https://arxiv.org/abs/2202.03036

Disclaimer: I have no relationships with the authors of [1] [2] [3], just in case.

**Summary Of The Paper:**

This paper presents a novel network architecture called Substructure-Atom Cross Attention (SACA) for molecular graphs that effectively combines Transformers and GNNs. Substructural patterns are important in molecular tasks, as we see in traditional chemoinformatics fingerprints like ECFP and fragmentation like BRICS/RECAP, but it is also widely known to be essentially hard for GNN to recognize substructure isomorphism. GNN is known to be upper bounded by 1-WL, whereas higher-order WL usually requires heavy computational cost. So explicitly incorporating substructural motifs can be a nice inductive bias to improve the prediction of GNNs in molecular tasks. The paper's simple idea is 1) prepare a motif vocabulary (e.g., MACCS keys), 2) for an input molecule, motif patterns detected from SMILES are fed as tokens to Transformers, 3) in parallel, the molecular graph is fed to GNN to get the node features representing local information around each node. 4) Then, GNN node features are used as keys & values in a Cross Attention layer with Transformer features as queries, 5) build a stack of Self Attention -> Cross Attention + a stack of Self Attention along with the CLS token for global output. The empirical studies demonstrated that this substructure-Transformer + molecule-GNN performed well in 11 downstream tasks after pretraining with a large ChEMBL + PubChem dataset.


**Summary Of The Review:**

This paper presents a novel network architecture called Substructure-Atom Cross Attention (SACA) for molecular graphs that effectively combines Transformers and GNNs. It is based on a simple idea of the use of predefined motifs (MACCS keys). The on-bit motifs are fed into Transformer as tokens (via embedding layers), while the input molecular graph is processed by a GNN and feedback to the Transformer through the Cross Attention layer. Even though I have several concerns, as described in the [Weaknesses] section, I liked the paper overall. It would always be nice to see that a simple idea worked. On the other hand, I believe that MACCS keys should be replaced by better ones to make inputs (almost) always valid.

---

> ### Author Response · Authors · 2022-11-23
> **Response to Reviewer Jvvo (1/3)**
>
> We deeply appreciate your insightful comments and constructive feedback. Here, we address the comments and questions mentioned.
>
> **1. Data leakage in evaluation from pretraining to downstream task.**
>
> We made sure that there were no overlapping molecules between pretraining and downstream datasets before the experiment. Therefore, we believe there is no data leakage from pretraining for each downstream task. We describe the pretraining dataset collection procedure. (1) We collected 2,271,376 unlabeled molecules from ChEMBL and PubChem databases. (2) We canonicalized the SMILES using RDKit to obtain the unique representation of molecules and remove all duplicated molecules in the pretraining dataset. (3) We retrieved all overlapping molecules between the pretraining and downstream datasets. (4) We removed the overlapping molecules from the pretraining dataset and kept the downstream dataset intact. Such a procedure results in 2,271,376 - 413,295 = 1,858,081 molecules in the pretraining dataset. Table 1 shows the statistics of overlapping molecules between each downstream dataset and the originally collected pretraining dataset. We add the details of the preprocessing steps in Section D of the Appendix.
>
> Table 1. Overlapping molecules between the original pretraining dataset and each downstream dataset.
>
>
> |                    | BBBP  | BACE | ToxCast | Tox21 | ClinTox | SIDER | FreeSolv | ESOL  | QM8   | QM7  | Lipo  |
> |--------------------|-------|------|---------|-------|---------|-------|----------|-------|-------|------|-------|
> | # mols             | 2039  | 1513 | 8576    | 7831  | 1478    | 1427  | 642      | 1128  | 21786 | 6830 | 4200  |
> | Overlapping # mols | 1029  | 3    | 8243    | 7105  | 631     | 956   | 459      | 826   | 869   | 529  | 1260  |
> | Ratio              | 50.5% | 0.2% | 96.1%   | 90.7% | 42.7%   | 67.0% | 71.5%    | 73.2% | 4.0%  | 7.7% | 30.0% |
>
>
> **2. If the input molecule doesn't have any of 166 structural keys, the input to the Transformer branch (the main branch) becomes empty, and this model doesn't make any sense. This would be one of the most well-known disadvantages to use predefined motifs (as in MACCS keys) rather than motifs occurring in the dataset (as in ECFP). So what was the average number of on-bit keys out of 166 motifs? This case would not be likely to happen?**
>
>
> We provide the statistics of MACCS keys (average, standard deviation, median, minimum and maximum number of MACCS keys) extracted for the pretraining dataset and downstream datasets in Table 2. We confirm that about 37 on-bit keys are extracted from the datasets on average, and **the minimum number of on-bit keys is at least 1 for every dataset**. We observe that no molecules have a 0 on-bit key - this is because MACCS keys include many fundamental substructures such as the methyl group (CH3), the existence of an oxygen atom, a nitrogen atom, etc. Therefore, most molecules have at least 1 MACCS key substructure.
>
> However, although we have not observed this in our datasets, there might be a corner case where no MACCS key is detected for a molecule. Even in that case, our model still injects the CLS token aggregating the node information through cross attention. In other words, without substructures, the CLS token would aggregate the node information from the GNN branch in the proposed network. In this case, our model may not benefit from the interaction between substructures and nodes, but it can still perform prediction via the GNN branch.
>
>
> Table 2. MACCS keys statistics for the pretraining and downstream dataset.
>
> | Dataset             | Average       | Median | Min | Max |
> |---------------------|---------------|--------|-----|-----|
> | Pretraining dataset | 52.12 ± 13.33 | 53     | 1   | 106 |
> | BBBP                | 46.03 ± 14.48 | 46     | 2   | 96  |
> | SIDER               | 46.61 ± 17.93 | 46     | 1   | 105 |
> | ClinTox             | 46.13 ± 16.54 | 46     | 2   | 92  |
> | BACE                | 61.04 ± 12.53 | 60     | 21  | 93  |
> | Tox21               | 32.73 ± 16.63 | 30     | 2   | 99  |
> | ToxCast             | 33.50 ± 17.30 | 31     | 2   | 101 |
> | FreeSolv            | 15.26 ± 9.78  | 13     | 1   | 62  |
> | ESOL                | 23.63 ± 15.52 | 19     | 1   | 76  |
> | Lipophilicity       | 51.61 ± 14.26 | 52     | 7   | 93  |
> | QM7                 | 18.75 ± 7.80  | 18     | 1   | 50  |
> | QM8                 | 23.00 ± 8.10  | 23     | 1   | 48  |

---

> > ### Author Response · Authors · 2022-11-23
> > **Response to Reviewer Jvvo (2/3)**
> >
> > **3. Another concern is about the use of MACCS keys. It is a set of 166 keys and is usually considered a bit weak to represent molecules in the entire ChEMBL and PubChem database. This is basically why PubChem  developed their own version of substructural keys called PubChem Fingerprint, which is an 881-bit-long structural key defined at https://pubchemdocs.ncbi.nlm.nih.gov/data-specification.**
> >
> > Thank you for suggesting an additional candidate for substructures. Following the suggestion, we conducted additional experiments with Pubchem Fingerprint and reported the result in Table 3 (also included in Appendix F). In general, MACCS keys show better performance on most of the datasets. It is interesting to see that the increased number of substructure vocabulary does not necessarily bring performance improvement. We speculate several reasons for this phenomenon. First, as there are many defined substructures, it needs more parameters to be learned and, thus, requires pretraining with much larger datasets. Second, Pubchem Fingerprint has some redundancy (e.g., many similar definitions for substructures), which is suboptimal for a data-driven approach. Although MACCS keys may have weak representation power, we think that the GNN branch used in our model encodes the whole molecular graph and complements the substructure information.
> >
> > **4. Also, it seems no problem if we just use data-driven motifs, by BRICS or ECFP, if the on bits are hashed into a fixed number of bits. Have you tested any other options than MACCS keys, particularly data-driven ones such as BRICS motifs or ECFP keys?**
> >
> > Yes, we have tested other data-driven motifs. Please note that we had included experiments using (1) ECFP fingerprint and (2) Tree decomposition in our original submission. During rebuttal, we conduct an additional experiment using Pubchem Fingerprint in the revision. Table 3 shows the results. While other substructures show competitive performance on most of the downstream tasks, MACCS keys still show the best performance on average.
> >
> > We highlight the differences between MACCS keys and the others (ECFP or Tree decomposition) and discuss the differences in performance as follows. First, ECFP or Tree decomposition does not utilize explicit functional groups. Particularly, ECFP extracts substructures based on a local neighborhood structure of nodes, which is similar to how GNN encodes node representations. As we use a GNN as a separate branch, the use of an ECFP fingerprint may not provide additional gain to the model.
> > Secondly, the possible number of substructures for data-driven motifs is much larger than those from MACCS Key. Specifically, ECFP encodes molecules into 1024 bits, which may require additional network parameters to learn the relationship between each substructure and node.
> >
> >
> > Table 3. Comparison between different substructures applied to our model.
> >
> > | Method              | BBBP      | BACE      | Tox21     | ToxCast   | FreeSolv | ESOL      | Lipo      | QM8        |
> > |---------------------|-----------|-----------|-----------|-----------|----------|-----------|-----------|------------|
> > | MACCS key           | **0.934** | 0.868     | **0.818** | **0.725** | 2.00     | **0.878** | **0.582** | **0.0140** |
> > | ECFP 4              | 0.925     | **0.869** | **0.818** | 0.716     | 2.52     | 0.900     | 0.596     | 0.0152     |
> > | ECFP 6              | 0.903     | 0.861     | **0.818** | 0.709     | 2.30     | 0.949     | 0.592     | 0.0151     |
> > | Tree Decomposition  | 0.925     | 0.848     | 0.769     | 0.715     | 2.37     | 0.885     | 0.614     | -          |
> > | Pubchem Fingerprint | 0.922     | 0.853     | 0.802     | 0.709     | **1.83** | 0.977     | 0.719     | -          |

---

> > > ### Author Response · Authors · 2022-11-23
> > > **Response to Reviewer Jvvo (3/3)**
> > >
> > >
> > > **5. The following three related methods [1] [2] [3] were not included in the empirical evaluation of the paper.**
> > >
> > > Thank you for suggesting additional baselines to compare with our model. In revision, we compare our model to the mentioned baselines (MGSSL [1], HMGNN [2], and SAT [3]). We evaluated MGSSL [1] with our MoleculeNet setting and included the results in Table 2 of the paper. For comparison against HMGNN [2] and SAT [3], we evaluate our model on two more datasets (i.e., ZINC and OGBG-MolHIV) following the evaluation setting used in [2], [3]. The results are shown in Table 3 and Table 4 in Section 4.2. The result shows that our model consistently outperforms MGSSL [1], HMGNN [2], and SAT [3].
> > >
> > > **6. Of particular interest, the "Structure-Aware Transformer (SAT)" [3] is based on Transformer + GNN like, Is this pattern "End-Concat"..? (First, motifs occurring in the input molecule are fed into GNNs, and their outputs are fed into Transformer)**
> > >
> > > We highlight the key difference between SAT and our model. In SAT, k-hop subgraphs are used as query and key, and nodes as value for attention calculation. Therefore, it computes subgraph-subgraph self attention and aggregates node features based on this attention score. The motivation is to increase discriminability as computing attention based on only node features cannot discriminate between two certain different nodes. On the other hand, our model uses molecular substructures as query, nodes as key and value. Therefore, we compute substructure-node cross attention and aggregate node features based on this attention score. We aim to learn the relation between semantic motifs (i.e. substructures) and atoms and to aggregate structurally important node features with more weights. We showed this effect in the cross attention map visualization described in Figure 7 in Section 4.4. We would like to clarify that End-Concat is not the same as SAT. End-Concat takes atoms as input, runs a vanilla transformer on them, and concatenates substructure features before the last linear layer. For further clarification, End-Concat is illustrated in Figure 11 in Appendix H.
> > >
> > >
> > > Reference
> > >
> > > [1] Zhang, Zaixi, et al. "Motif-based graph self-supervised learning for molecular property prediction."  (NeurIPS 2021)
> > >
> > > [2] Yu, Zhaoning, and Hongyang Gao. "Molecular representation learning via heterogeneous motif graph neural networks." (ICML2022)
> > >
> > > [3] Chen, Dexiong, Leslie O’Bray, and Karsten Borgwardt. "Structure-aware transformer for graph representation learning." (ICML2022)

---

> ### Comment · Reviewer_Jvvo · 2022-11-30
> **I acknowledge that I have read the responses and updates**
>
> Thank you for the clarification. The additional info and updates were clear and informative.
>
> When the input molecule doesn't have any of the MACCS substructural keys, the proposed concept of Substructure-Atom Cross Attention would not take any advantage of substructural motif information, but I also understood that it still performs the prediction by feeding the molecular graph information from GNN branch to the CLS token passing, and it causes at least no harm to consider the Transformer branch based on the MACCS keys.
>
> But at the same time, out of 166 bits, FreeSolv, ESOL, QM7 have only 13-19 on-bits at median, and it would be still a bit unconvincing that feeding this information via cross attention is a major cause of the observed performance improvement. If that is the case, the MACCS keys are only 166 bits, i.e., of fixed length, and thus, (instead of the Transformer branch) just feeding this 166-dimensional binary vector to MLP (or Transformer Encoder) and considering cross attention between MLP and GNN might also work. It should be informative to know the existence as well as the **absence** of the MACCS keys.

---

> > ### Author Response · Authors · 2022-12-11
> > **Response to Reviewer Jvvo**
> >
> > We deeply appreciate your further feedback and additional discussion points. Here, we address the comments and questions mentioned.
> >
> > **1. But at the same time, out of 166 bits, FreeSolv, ESOL, QM7 have only 13-19 on-bits at median, and it would be still a bit unconvincing that feeding this information via cross attention is a major cause of the observed performance improvement.**
> >
> > Thank you for pointing out a good discussion point. One main reason for the low number of on-bit MACCS keys for the mentioned datasets is that the molecule size for those datasets is relatively small. In Table 1, we present the statistics of molecule size (number of atoms) for each downstream dataset. It shows that the mentioned datasets (FreeSolv, ESOL, QM7) have small molecule sizes, with a median number of 7-12. Correspondingly, the detected numbers of MACCS keys for the datasets are small. Also, the **rank correlation between the median number of MACCS keys and the median number of molecule sizes for each dataset is 0.961**, showing a high correlation between the two. Therefore, the small number of on-bit MACCS keys is not a deficiency of information but rather a natural consequence of the small molecule size.
> >
> > We further visualize the t-SNE embeddings of MACCS keys with labels. They are shown in Figure 1, 2, 3 at the anonymized link [1]. Although the clustering is not perfect, we can observe that some molecules are clustered with the corresponding labels. Considering the fact that the t-SNE embeddings are not from a trained model, MACCS keys, even a small number, have meaningful information to predict molecular properties for the mentioned dataset.
> >
> > Table 1. Statistics of molecule size (number of atoms) for each downstream dataset.
> > |    Dataset    |      Average     | Median |  Min  |   Max  |
> > |:-------------:|:----------------:|:------:|:-----:|:------:|
> > |      BBBP     |   24.06 ± 10.59  |   23   |   2   |   132  |
> > |     SIDER     |   33.64 ± 47.90  |   25   |   1   |   492  |
> > |    ClinTox    |   26.16 ± 15.61  |  23.0  |   1   |   136  |
> > |      BACE     |   34.09 ± 8.52   |   33   |   10  |   97   |
> > |     Tox21     |   18.57 ± 11.35  |   16   |   1   |   132  |
> > |    ToxCast    |   18.78 ± 11.58  |   16   |   2   |   124  |
> > |  **FreeSolv** |  **8.72 ± 4.19** |  **8** | **1** | **24** |
> > |    **ESOL**   | **13.29 ± 6.88** | **12** | **1** | **55** |
> > | Lipophilicity |   27.04 ± 7.46   |   27   |   7   |   115  |
> > |    **QM7**    |  **6.79 ± 0.53** |  **7** | **1** |  **7** |
> > |      QM8      |    7.77 ± 0.55   |    8   |   1   |    8   |
> >
> > **2. If that is the case, the MACCS keys are only 166 bits, i.e., of fixed length, and thus, (instead of the Transformer branch) just feeding this 166-dimensional binary vector to MLP (or Transformer Encoder) and considering cross attention between MLP and GNN might also work. It should be informative to know the existence as well as the absence of the MACCS keys.**
> >
> > Thank you for suggesting a further experiment to figure out how the information on the absence of the MACCS keys is effective. Following the suggestion, we designed an additional model as follows. When we embed substructure tokens, we use two separate embedding layers: one for on-bit keys and the other for non-on-bit keys to distinguish the presence and absence of the MACCS keys. An illustration of the model is shown in Figure 4 at the attached link [1]. Therefore, the information on the absence as well as the presence of the MACCS keys is used with the fixed 166 input tokens. We are currently running the experiments and will report the results in the next few days.
> >
> > [1] Link for figures: https://anonymous.4open.science/r/iclr_discussion-996E/README.md

---

> > > ### Author Response · Authors · 2022-12-12
> > > **Response to Reviewer Jvvo**
> > >
> > > **Experimental results for the absence-considered model.**
> > >
> > > We designed an additional model that considers absent tokens, as shown in Figure 4 of the anonymized link [1]. Similar to our original model, we pretrained the additional model on 1.8M molecules collected from ChEMBL and PubChem databases and then finetuned on each downstream dataset. We present the experimental results in Table 1. As shown in the results, there is almost no difference between the two models in performance, and their average rankings are almost the same.
> > >
> > > We had various discussions about the reasons. For example, the cross attention mechanism between not-on-bit tokens and existing atoms in a molecule is unnatural in our model design, or the not-on-bit keys information might be already considered implicitly by pretraining the model with on-bit keys on large datasets. In the discussion period, we could not reach a definite conclusion. However, we agree that the absence of MACCS keys can be informative, and we leave further exploration of its usage as future work.
> > >
> > >
> > > Table 1. Comparison between our model and absence-considered model.
> > > |                    |  BBBP     | SIDER     | Clintox   | BACE      | Tox21     | ToxCast   | FreeSolv | ESOL      | Lipo      |
> > > |--------------------|-----------|-----------|-----------|-----------|-----------|-----------|----------|-----------|-----------|
> > > | Ours               | 0.934     | **0.646** | **0.935** | **0.877** | **0.829** | 0.730      | 1.75     | **0.822** | 0.575     |
> > > | Absence-considered | **0.937** | 0.644     | 0.932     | 0.867     | 0.822     | **0.733** | **1.71** | 0.836     | **0.573** |

---

### Author Response · Authors · 2022-11-23
**Dear reviewers and AC**

We would like to thank all the reviewers for their insightful comments and constructive feedback. In a general comment, we summarize the additional details and further experiments with the corresponding sections in the revised manuscript.
We provide more specific responses to each reviewer below. We look forward to having additional feedback and discussion.

We have addressed common concerns about our experiments and added the details as follows.
- We add more detailed preprocessing procedures to clarify there is **no data leakage** from pretraining for downstream datasets in Appendix D [R1, R4].
- We add more information about the existence of positional encodings and hyperparameters for architecture search [R3].
- We provide additional MACCS keys statistics (e.g, average, median, minimum, etc) for pretraining and each downstream dataset in Table 8 of Appendix C [R1].
- We illustrate four variants of Transformer-based models that utilize substructures and describe the key differences between them to demonstrate the novelty of our model in  Appendix H [R2, R4, R5].
- We clarify the reason why we consider our model complexity O(n) [R2, R5].

We have conducted additional experiments in response to the suggestions.
- We compare our model with more baselines on [1], [4], [5] in Table 2 of Section 4.2 [R1, R4, R5].
- We report performances on two additional datasets (OGBG-MolHIV, ZINC) to compare with [2], [3] in Table 3 and 4 of Section 4.2 [R1].
- We report the performance with the Pubchem fingerprint in Table 13 in Appendix F [R1].
- We report the performance of direct ensemble models of GROVER and GIN to show the technical contribution of our model [R4].


Reference

[1] Zhang, Zaixi, et al. "Motif-based graph self-supervised learning for molecular property prediction." (NeurIPS 2021)

[2] Yu, Zhaoning, and Hongyang Gao. "Molecular representation learning via heterogeneous motif graph neural networks." (ICML 2022)

[3] Chen, Dexiong, Leslie O’Bray, and Karsten Borgwardt. "Structure-aware transformer for graph representation learning." (ICML 2022)

[4] Wu, Fang, et al. "3D-Transformer: Molecular representation with transformer in 3D space." arXiv preprint arXiv:2110.01191.

[5] Liu, Shengchao, et al. "Pre-training molecular graph representation with 3d geometry." (ICLR 2022)

---

### Decision · Program_Chairs · 2023-01-20

**Decision:**

Reject

**Justification For Why Not Higher Score:**

There are too many open questions, some of which concern severe conceptual problems.

**Justification For Why Not Lower Score:**

N/A

**Metareview: Summary, Strengths And Weaknesses:**

For this paper, we have four reviews with rather neutral scores. These four reviewers mentioned a couple of strong points, but also some problems, such as
- the use of predefined substructures (the 166 substructures of MACCS keys) and situations in which an  input molecule doesn't have any of these. At the end of the discussion phase one  reviewer (who initially assigned a slightly positive score) expressed that she/he was convinced by the answers provided be the authors, and to be more inclined to the negative after the discussion phase.
-  the potential risk of data leakage in evaluations. Although the authors explained in detail their procedure of excluding any duplicated molecules in the external training set, I still think that this answer does not fully address potential leakage problems, since there might still be extremely similar (but not completely identical) molecules left.
-  missing novelty. I also think that the mentioned conceptual similarity to MolFormer [1] can be somewhat problematic. Although the authors present some comparisons in the revision, some question marks remain (because it is hard to see truly substantial conceptual differences between the approaches).
-  issues with the statements about computational complexity. It seems that some of these issues have be confirmed by the authors added some further details (which, however, do not look as impressive as the initial claim of linear complexity).
-  unclear  /lacking technical contributions.

I think it is important to note that none of these four initially "neutral" reviewers could be convinced to assign a clearly positive score (if any, there was even a tendency to the negative side).

Then there is another (initially) clearly positive review. After the discussion phase, however, this reviewer also saw some potential problems (such as unclear novelty) and indicated that the original score was too high (still on the "positive side, though less so than originally").

After going over all reviews and discussions again, I come to the conclusion that for this paper the negative aspects dominate the positive ones. Even after the rebuttal, there seem to be too many open questions, and some of these are truly fundamental questions regarding the conceptual basis / novelty / scientific depth of this work. Therefore, I vote for rejection.